# Analysis of a shark reveals ancient, Wnt-dependent, habenular asymmetries in vertebrates

Maxence Lanoizelet [1,14], Léo Michel [1,14], Ronan Lagadec [1,14], Hélène Mayeur[1,14], Lucile Guichard [1,14], Valentin Logeux[2], Dany Séverac[3], Kyle Martin[4], Christophe Klopp [5], Sylvain Marcellini [6], Héctor Castillo[6], Nicolas Pollet [7], Eva Candal [8], Mélanie Debiais-Thibaud [9], Catherine Boisvert[10], Bernard Billoud [11], Michael Schubert [12], Patrick Blader [13] & Sylvie Mazan [1] ✉

The mode of evolution of left-right asymmetries in the vertebrate habenulae remains largely unknown. Using a transcriptomic approach, we show that in a cartilaginous fish, the catshark *Scyliorhinus canicula*, habenulae exhibit marked asymmetries, in both their medial and lateral components. Comparisons across vertebrates suggest that those identified in lateral habenulae reflect an ancestral gnathostome trait, partially conserved in lampreys, and independently lost in tetrapods and neopterygians. Asymmetry formation involves distinct mechanisms in the catshark lateral and medial habenulae. Medial habenulae are submitted to a marked, asymmetric temporal regulation of neurogenesis, undetectable in their lateral counterparts. Conversely, asymmetry formation in lateral habenulae results from asymmetric choices of neuronal identity in post-mitotic progenitors, a regulation dependent on the repression of Wnt signaling by Nodal on the left. Based on comparisons with the mouse and the zebrafish, we propose that habenular asymmetry formation involves a recurrent developmental logic across vertebrates, which relies on conserved, temporally regulated genetic programs sequentially shaping choices of neuronal identity on both sides and asymmetrically modified by Wnt activity.

Knowledge on how cerebral structures diversify during evolution is crucial for understanding how animals adapt their behavioral strategies to varied environmental challenges. The vertebrate habenula presents unique characteristics to address this general question. This bilateral epithalamic structure forms a relay in conserved neuronal circuits connecting various forebrain areas to midbrain and brainstem nuclei. It appears as a key node, which integrates information from multiple sources (sensory organs, corticolimbic areas), and regulates a variety of adaptive behavioral, cognitive and emotional responses[1–6]. The neuronal basis for these regulations is a field of intense

investigations in the mouse and the zebrafish, but their conservation remains largely unexplored in other species and different ecological contexts. Habenular territories have also been implicated in human pathological conditions, such as depression or drug addiction[7,8], and knowledge transfers from animal models to humans require a robust knowledge of cross-species conservations and divergences.

A remarkable feature of habenulae is that in many vertebrates, they display asymmetries between their left and their right sides. Their biological significance has started to emerge in the zebrafish, the reference model for their analysis[9]. In this species, habenular

asymmetries result in a differential integration of sensory cues between the right and the left habenulae, and they impact important adaptive responses, such as exploratory and food-seeking behaviors, light preference or responses to fear[10–13]. Besides the evolutionary significance of these functions, this trait is highly significant from an evolutionary developmental biology perspective, as the generation of neuronal diversity both during ontogeny, between the two sides of a same, functional, structure, and during evolution, depending on ecological contexts, paves the way for explorations of connections between these two levels.

To date, the origin and the mode of evolution of habenular asymmetries remain unclear. Habenular asymmetries in size, cellular organization, and connectivity pattern, have been reported in all major vertebrate taxa including mammals, but histological comparisons suggest considerable variations in their nature and degree[14–16]. Detailed characterizations have thus far only been carried out in a limited number of species, primarily the zebrafish and the mouse, as well as, to a lesser extent, the lamprey.

In the zebrafish, habenulae are subdivided into a dorsal and a ventral component, a partitioning conserved in gnathostomes (jawed vertebrates)[16–19], albeit with a medial and lateral organization in tetrapods. Asymmetries in the zebrafish are restricted to the dorsal habenula, and consist of different relative proportions between two nuclei, respectively occupying lateral (dHbl) and medial locations (dHbm), on both the left and the right sides[20–22]. Comparisons across teleosts reveal marked variations of this asymmetry pattern, with complex presence/absence patterns of other asymmetric traits, suggestive of a rapid evolutionary drift[23–25]. Various, more subtle asymmetries in size, gene expression, neuronal activity, and projection pattern have been reported in mice and humans both in lateral and medial habenulae, but with significant inter-individual differences and without obvious evolutionary relationships to those described in teleosts[15,26,27].

In the river lamprey, a member of cyclostomes (or jawless vertebrates), which diverged from gnathostomes, their sister group in vertebrates, more than 500 million years ago (Mya)[28], analyzes of afferent and efferent projections highlighted the conservation of the bipartite organization of habenulae. However, they suggested different relative locations of territories homologous to the dorsal/medial and ventral/lateral components of gnathostomes, and a highly distinctive asymmetry pattern. Putative lamprey homologs of the ventral/lateral habenula of gnathostomes indeed map exclusively to the right habenula, while those of the dorsal/medial habenula are distributed between both sides, comprising the entire left habenula, and a smaller, right-restricted, medial cell nucleus[29,30].

Not only the degree and the nature of habenular asymmetries, but also the mechanisms underlying their formation, vary across species. Analyzes in the zebrafish have demonstrated a pivotal role for Wnt signaling. Activity of the pathway is submitted to a finely tuned temporal control during habenular development[31–33], and its repression on the left side via an interaction with the adjacent parapineal during a discrete time window is required for the elaboration of habenula asymmetries[20,34,35]. The underlying cellular mechanism has been elucidated: it relies on an asymmetric temporal control of neurogenesis, promoting the choice of a right-prevailing neuronal identity (dHbm) in the right dorsal habenula[32,36].

Analyzes of a shark and a lamprey, both endowed with marked habenular asymmetries, reveal a major difference with the zebrafish. While the involvement of Wnt signaling remains an opened question in these species, they both require an early left-restricted diencephalic activity of Nodal, dispensable in the zebrafish, for habenular asymmetry formation[37]. In view of these variations, it is still unclear whether habenular asymmetries arose several times independently in the different vertebrate lineages, possibly due to common developmental constraints, or whether they diversified from an ancestral pattern, differentially modified in individual lineages. To clarify the origin and the mode of evolution of habenular asymmetries across vertebrates, we have focused on the catshark *Scyliorhinus canicula*, a member of chondrichthyans (cartilaginous fishes)[38], using an unbiased transcriptomic characterization and systematic phylogenetic comparisons.

Here we show that the highly asymmetric habenular organization observed in the catshark has retained ancient vertebrate asymmetries, which were independently lost in the mouse and the zebrafish. We also show an essential role of Wnt signaling in their formation in lateral habenulae, in line with a conserved Wnt-dependent developmental regulatory logic for asymmetry formation accounting for the recurrence and evolvability of habenular asymmetries across vertebrates.

## Results

### Asymmetrically expressed genes in developing catshark habenulae

To obtain an unbiased characterization of molecular asymmetries in catshark habenulae, we carried out a transcriptomic comparison between left and right developing habenulae. We focused on stage 31, when a large fraction of the structure is differentiated but asymmetric pools of neural progenitors persist[39]. Statistical analysis of 362 million read pairs obtained from three replicates of left versus right habenulae explants led to the identification of 614 differentially expressed gene models between the left and the right sides (373 left-enriched and 241 right-enriched), of which 538 were annotated as protein-coding (Fig.1a; Supplementary Data 1). Functional annotation of left- and right-enriched gene lists showed that they share an over-representation of GO (Gene Ontology) terms related to neurogenesis, neuronal differentiation, formation of neuronal projections and trans-synaptic signaling, as expected for a differentiating brain structure (Supplementary Fig. 1a,b; Supplementary Data 1). GO terms selectively over-represented in either one of these two lists were also detected. They include terms related to organismal responses known to be regulated by habenulae, such as the perception of pain and behavioral fear responses, and to signaling pathways known to be active in habenulae (Wnt, opioid receptor, and ionotropic glutamate receptor signaling) (Supplementary Fig. 1c-d). Altogether, these data suggest differential functional specializations of the catshark left and right habenulae.

### Asymmetric lateral to medial organization of catshark habenulae

To validate asymmetries identified by the transcriptomic analysis and to characterize their spatial organization, we performed chromogenic in situ hybridization (ISH) on sections of stage 31 habenulae for a total of 39 genes, selected from the lists of asymmetrically expressed genes (Fig.1a). Regionalized signals were observed for most of these genes, with three of them showing restricted expressions in territories consisting of neural progenitors (Supplementary Fig. 2h)[39] and twenty-nine in differentiating subdomains of the habenulae, all consistent with the expression lateralities predicted by the transcriptomic analysis (Supplementary Fig. 2b–g). Most of the twenty-nine genes can be classified into five broad categories, each characterized by largely overlapping expression profiles restricted to (i) a left lateral territory (Supplementary Fig. 2b), (ii) a right lateral territory (Supplementary Fig. 2f), (iii) a broad bilateral medial territory, larger on the left than on the right (Supplementary Fig. 2d), (iv) a left-restricted subdomain of this medial territory (Supplementary Fig. 2c), and (v) a right-enriched subdomain of the medial territory (Supplementary Fig. 2e).

More detailed analyzes focusing on markers of each of these territories (*ScSox1*, *ScProx1*, *ScKctd12b*, *ScPde1a*, and *ScEnpp2*; Fig.1; Supplementary Figs. 3 and 4) show an organization of stage 31 catshark habenulae into three broad complementary subdomains, two lateral ones, referred to as Left- and Right-LHb hereafter, expressing *ScSox1* on the left and *ScProx1* on the right, and a broad bilateral *ScKctd12b*-positive medial one, referred to as MHb, including a previously

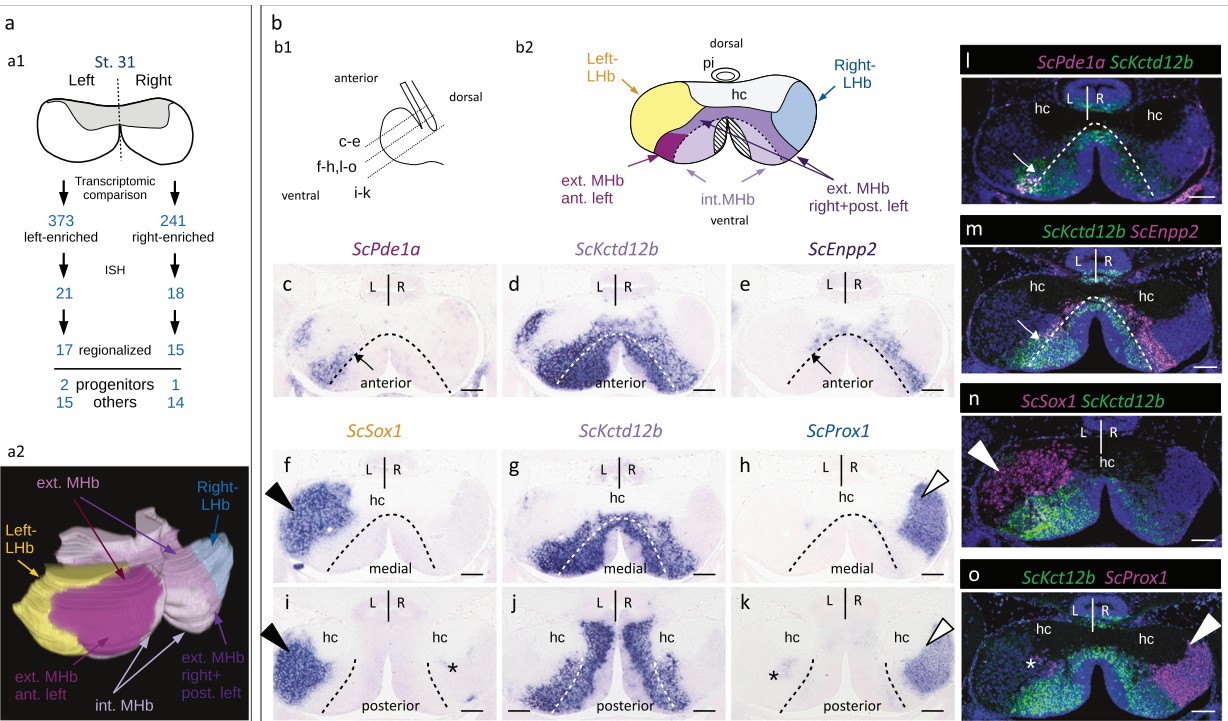

**Fig. 1 | Developing catshark habenulae harbor major asymmetries both in lateral and medial compartments. a** Schemes showing the experimental strategy for the characterization of molecular asymmetries in catshark habenulae (**a1**) and the resulting 3D organization at stage 31 (**a2**). Anterior is to the left, dorsal to the top. **b** Schemes showing a left lateral view of catshark stage 31 habenulae, with section planes and levels indicated by dotted lines (**b1**), and the subdomain organization observed on a transverse section at a medial level (**b2**). **c**–**o** Transverse sections (dorsal to the top of each panel) after in situ hybridization (ISH) with probes for *ScPde1a* (**c**), *ScKctd12b* (**d,g,j**), *ScEnpp2* (**e**), *ScSox1* (**f,i**) and *ScProx1* (**h,k**), and after fluorescent double ISH with probes for *ScPde1a/ScKctd12b* (**l**), *ScKctd12b/ScEnpp2* (**m**), *ScSox1/ScKctd12b* (**n**) and *ScKctd12b/ScProx1* (**o**). For fluorescent double ISH, signals for *ScKctd12b* are shown in green and those for *ScPde1a*, *ScEnpp2*, *ScSox1* and *ScProx1* in magenta. Sections (**c**–**k**) were obtained from the same embryo. Black and white arrowheads in (**f,h,i,k**) respectively point to major lateral territories of *ScSox1*

and *ScProx1*. White arrowheads point to major *ScSox1* and *ScProx1* major lateral territories in (**n**) and (**o**) respectively. Asterisks in (**i,k,o**) show contra-lateral minor posterior territories. Dotted lines in (**c**–**m**) delimit external and internal subdomains of the medial habenula, as inferred from the inner boundaries of *ScPde1a* and *ScEnpp2* territories. Thin arrows in (**c,e,l,m**) point to the boundary between the complementary territories of *ScPde1a* and *ScEnpp2* within the external MHb subdomain. Color code in (**a2**): yellow, Left-LHb; light purple, internal MHb; magenta, anterior, left-restricted component of external MHb; dark purple, right, plus posterior-left components of external MHb; blue, Right-LHb; hatched, pseudo-stratified neuroepithelium containing neural progenitors. Color code in (**b2**): same as in (**a2**); hatched, neural progenitors. The same ISH profiles were consistently obtained for each gene on at least five different specimens. Abbreviations: ant., anterior; post., posterior; ext., external; int., internal; MHb, medial habenula; LHb, lateral habenula; hc, habenular commissure; pi, pineal stalk; L, left; R, right; st., stage. Scale bar = 100 μm.

described left-restricted anterior component[37] (Fig.1a2,b2,d,f–k,n,o; Supplementary Fig. 3b,d,f; Supplementary Fig. 4b,d,f). *ScSox1* and *ScProx1* expression patterns are confined, respectively, to the Left-LHB and the Right-LHb, except for a small contralateral neuronal population at the posterior-most level of the habenulae (Fig.1i,k; Supplementary Fig. 3b5,f4 and Supplementary Fig. 4b4,f3). The bilateral MHb is further partitioned into an internal component and an external one, adjacent to Left-LHb and Right-LHb (Fig.1c–e; Supplementary Figs. 3c–e,4c–e). The external MHb component is organized into two discrete abutting territories, an anterior, left-restricted *ScPde1*-positive territory (Fig.1c,l; Supplementary Figs. 3c,4c), and a *ScEnpp2*-positive territory, occupying the whole external part of the MHb on the right and its posterior part, complementary to the *ScPde1a* domain, on the left (Fig.1e,m; Supplementary Figs. 3e,4e). To assess whether these asymmetries reflect transient ones in the differentiating habenulae or definitive neuronal identities of the functional organ, we investigated their maintenance in feeding juveniles collected after exhaustion of their yolk reserves (Supplementary Fig. 5). The broad expression characteristics and asymmetries of *ScSox1*, *ScPde1a*, *ScKctd12b*, *ScEnpp2*, and *ScProx1* are very similar to those observed at stage 31 (Supplementary Fig. 5b-k), except for an expansion of the *ScEnpp2* signal to the internal component of MHb (Supplementary Fig. 5e). Altogether, these results highlight a highly asymmetric organization of catshark habenulae, in both lateral and medial subdomains.

## Variable habenular asymmetries between the catshark, the mouse and the zebrafish

To characterize the level of conservation of habenular asymmetry and subdomain organization across gnathostomes, we initially focused on comparisons of the catshark with the mouse and the zebrafish. For this analysis, we systematically surveyed previously published habenular expression of mouse and zebrafish orthologs of markers of the five major habenular territories identified in the catshark, including *Kctd8/12a/12b* paralogous genes (Supplementary Table 1; Supplementary Fig. 6).

In the mouse, this analysis was based on profiles published in the habenulae[40–43], including systematic searches in the Allen Brain Atlas[44]. Most mouse orthologs of catshark MHb markers are expressed in the mouse medial habenula, including *Spon1* and *Trhde*, restricted to the anterior left external component of the MHb in the catshark and to a lateral subdomain of the medial habenula in the mouse (Supplementary Table 1; Supplementary Fig. 6a–d,l). Two of the four gene signatures of the catshark Left-LHb (*Ntng2* and *Pcdh17*) share bilateral expression in mouse lateral habenulae, with additional signals in the medial habenula (*Ntng2*) and in the paraventricular nucleus of the thalamus (PVT), adjacent and ventral to the lateral habenulae (*Pcdh17*) (Supplementary Fig. 6e,f). A highly specific lateral habenula expression was also observed for a *Sox1* reporter in *Sox1-LacZ* knock-in mice[41]. Mouse orthologs of catshark Right-LHb markers show various

expression profiles spanning the ventral medial habenulae, the lateral habenulae and adjacent thalamic nuclei (Supplementary Fig. 6g,k), with four of them sharing a highly specific expression territory in the PVT (*Rerg, Rora, Prox1* and *Stxbp6* (Supplementary Fig. 6g–i,k).

For comparisons with the zebrafish, we examined the presence of orthologs of catshark territory markers in gene signatures of single-cell RNA-seq gene clusters mapped to larval or adult habenulae[19], in addition to previously published ISH data[20–22,45] (Supplementary Table 1). Orthologs of three asymmetrically expressed catshark MHb markers (*Spon1, Kctd8,* and *Kctd12a*) display asymmetric expressions in the zebrafish dHb, albeit without consistent conservation of asymmetry laterality between the two species. Conversely, with one exception (*prox1a*, part of a dorsal and right-enriched cluster*)*, zebrafish orthologs of catshark LHb markers (*Sox1, Ak5, Kiss1, Rerg, Prkcq,* and *Gng14*) are identified as gene signatures of two bilaterally symmetric ventral cell clusters of zebrafish larval habenula, Hb11 for zebrafish *sox1a/b*, Hb15 (or related adult clusters) for the other five genes.

Taken together, these data support the conservation in the catshark of the subdivision of habenulae into two components (dorsal/medial and ventral/lateral). Intriguingly, these results further suggest similarities between the catshark Left-LHb and the mouse left and right lateral habenulae, and between the catshark Right-LHb and a major zebrafish ventral, bilateral cell cluster.

## Habenular asymmetry conservations across jawed vertebrates

The similarities of catshark Left- and Right-LHb respectively with the mouse lateral and the zebrafish ventral habenulae may reflect convergences or taxon-specific diversifications of lateral/ventral habenulae, unrelated to ancestral traits of habenular organization. To test their evolutionary relevance, we used a phylogenetic approach, aimed at reconstructing habenular organization at major gnathostome nodes, taking the catshark as reference. Our species sampling included representatives of the three main gnathostome phyla: chondrichthyans, actinopterygians (ray-finned fishes) and sarcopterygians (lobe-finned fishes and tetrapods). We selected five species for their phylogenetic position within these phyla: the elephant shark *Callorhinchus milii* (chondrichthyan, member of holocephalans, the sister group of elasmobranchs[46]), two non-teleostean actinopterygians, the reedfish *Erpetoichthys calabaricus* (member of polypterids, the earliest diverging lineage in actinopterygians[47]) and the spotted gar *Lepisosteus oculatus* (member of holosteans, which together with teleosts, their sister group, form neopterygians[48]), and two sarcopterygians, the African lungfish *Protopterus annectens* (member of lungfishes, the closest living relatives to tetrapods[49]) and the Western clawed frog, *Xenopus tropicalis*. In each one of these species, we carried out ISH analyzes on transverse sections of habenulae at stages when specimens were available (elephant shark: pre-hatching embryos; reedfish, spotted gar, lungfish: juveniles; frog: stage NF-66 tadpoles), using probes for orthologs of markers of the catshark Left-LHb (*Sox1, Ntng2,* and *Pcdh17*), MHb (*Kctd8, Kctd12a,* and *Kctd12b*) and Right-LHb (*Prox1* and *Kiss1*) (Fig.2; Supplementary Fig. 7–11).

Specific habenular expression was observed in all cases, except for *Kctd12a* in the lungfish, and *Kiss1* in the frog, which yielded no detectable ISH signals. In all species analyzed, profiles of members of the *Kctd8/12a/12b* family vary between paralogs and species, but at least one paralogue always defines a medial or dorsal habenular territory (Fig. 2g–l; Supplementary Fig. 7e–g; Supplementary Fig. 8f,i,j; Supplementary Fig. 9d–f; Supplementary Fig. 10e,f; Supplementary Fig. 11f,h–j). With a few exceptions (such as the expansion of elephant shark *Kiss1* and *Pcdh17*, reedfish *Prox1*, and spotted gar *Ntng2* to various MHb/dHb subterritories), orthologs of catshark Left- and Right-LHb markers are expressed within lateral or ventral territories (Fig.2b–f,n–r; Supplementary Fig. 7b–d,h,i; Supplementary Fig. 8b–e,g,h; Supplementary Fig. 9b,c,g–k; Supplementary Fig. 10b–d,g,h; Supplementary Fig. 11b–e,g). However, the labeled territories and asymmetry patterns

markedly differ between species. As in the catshark, the elephant shark and the reedfish show left- and right-restricted lateral/ventral territories, respectively co-expressing *Sox1/Pcdh17/Ntng2* and *Prox1/Kiss1* (Fig. 2b,c,n,o; Supplementary Fig. 7b–d,h,i; Supplementary Fig. 8b–e,g,h). Similarly, in the lungfish, *Sox1, Pcdh17,* and *Ntng2* share a strong left-restricted signal, adjacent to *Kctd8/12b* positive territories (Fig. 2e; Supplementary Fig. 10b–d). A more discrete lateral cell population positive for both *Prox1* and *Kiss1* is observable on the right side, in a zone devoid of *Kctd8/12* expression, adjacent to the habenular commissure (Fig. 2q; Supplementary Fig. 10g,h). In contrast, expression of orthologs of both catshark Left- and Right-LHb markers are bilateral in the spotted gar and the frog, albeit with slightly different relative locations between the two species. In the spotted gar, *Pcdh17/Sox1/Ntng2* share an anterior expression territory, complementary to the *Prox1/Kiss1*-positive one within the left and right ventral habenulae (Fig. 2d,p; Supplementary Fig. 9b,c,g–k). In the frog, their expression is also observed in both the left and right habenulae, adjacent and lateral to the domain of *Kctd8/12b* (Fig. 2f; Supplementary Fig. 11b–e), with *Prox1* being expressed at more ventral and posterior levels (Fig. 2r; Supplementary Fig. 11g).

## Nodal-dependent left repression of Wnt signaling in catshark developing habenulae

To gain insight into the mechanisms controlling the formation of asymmetries in the lateral/ventral habenulae of chondrichthyans, lungfishes, and polypterids, we focused on canonical Wnt signaling in the catshark, a model amenable to experimental approaches during development. Expressions of *ScLef1* and *ScTcf7l2*, which encode effectors of the canonical Wnt pathway, showed statistically significant right enrichments in our transcriptomic analysis (*q*-value = 2.2E-03 and 3.0E-03 respectively) (Supplementary Data 1). We analyzed the dynamics of Wnt activity during habenula formation, by assessing the nuclear distribution of β-catenin on histological sections, starting from the earliest appearance of habenular evaginations, at stage 26, and until stage 31 (Fig. 3; Supplementary Fig. 12).

At stage 26, β-catenin is broadly distributed in growing habenular evaginations, but a nuclear signal is only observed in some HuC/D-positive cells, which form a discrete lateral cell population of differentiating neurons, transiently larger on the left than on the right side at this stage[39] (Fig. 3a,b,a1–a4,b1–b4). A broad bilateral cytoplasmic signal is maintained at stage 28, albeit without clear nuclear restriction (Fig.3c1–c4). A marked transition is observed at stage 29, with a nuclear accumulation of β-catenin restricted to the lateral right habenula and a withdrawal of the cytoplasmic signal in the lateral left habenula (Fig. 3d1–d4). This asymmetry is maintained at stage 31, with a high proportion of positive nuclei in the *Prox1*-expressing Right-LHb and a complete absence of signal (neither nuclear, nor cytoplasmic) in the *Sox1*-positive Left-LHb (Fig. 3e, 3f; compare Fig. 3e1,e2 with Fig. 3e3,e4). In the *Kctd12b*-positive MHb (Supplementary Fig. 12a,c,e), contrary to to the right lateral habenula, the β-catenin signal is excluded from nuclei in all areas examined (see MHb signals in Supplementary Fig. 12b,d,f,b1–b2,d1–d8). However, a heterogeneity is observed, with cytoplasmic β-catenin expressions detected in all MHb subdomains except in the anterior left-restricted *ScPde1a*-positive external component (Supplementary Fig. 12b,d; compare Supplementary Fig. 12d1,d2 to d3,d4, and d5,d6 to d7,d8).

The restriction of Wnt signaling activity to the right catshark habenula at stages 29-31 is reminiscent of that reported in the zebrafish, which is controlled by a parapineal-mediated repression on the left[32,35]. In the absence of a parapineal in the catshark, we assessed the dependence of this asymmetry on Nodal signaling, by inhibiting the pathway during its window of left-restricted activity shortly after neural tube closure, as previously described[37] (Fig. 3g,h). While control embryos exhibit the expected restriction of nuclear β-catenin to the Right-LHb (Fig. 3g1–g4), *in ovo* injection of the Nodal/Activin

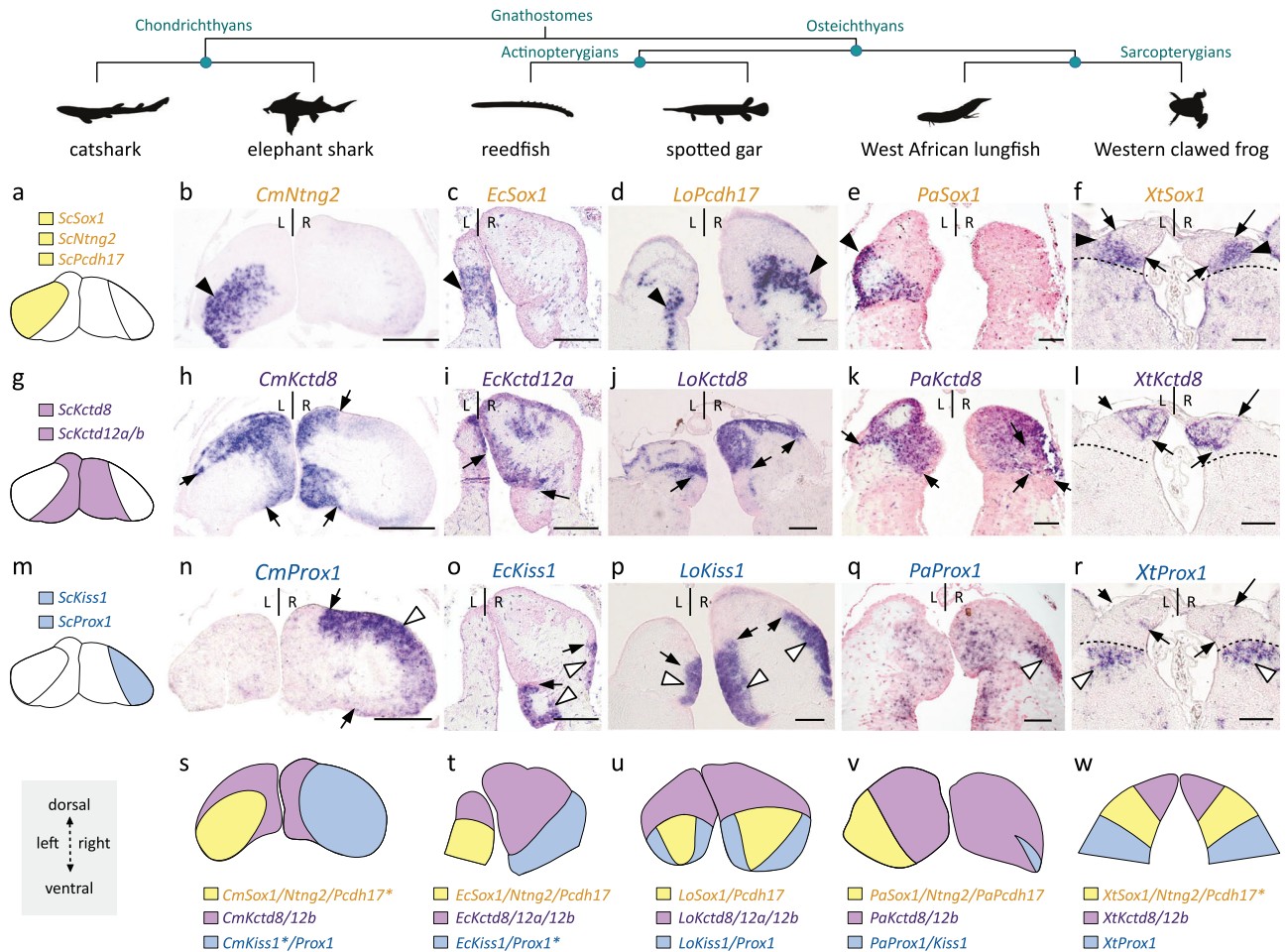

**Fig. 2 | Asymmetries related to those observed in catshark lateral habenulae are present in a lungfish and a polypterid, but undetectable in members of tetrapods and neopterygians. a,g,m** Schemes showing the asymmetric organization of the catshark habenulae, with the reference signature markers of Left-LHb (**a**, yellow), MHb (**g**, purple) and Right-LHb (**m**, blue) considered for cross-species comparisons. **b–f,h–l,n–r** Transverse sections of habenulae in the elephant shark (stage 36 embryo) (**b,h,n**), the reedfish (juvenile) (**c,i,o**), the spotted gar (juvenile) (**d,j,p**), the African lungfish (juvenile) (**e,k,q**) and the Western clawed frog (stage NF-66 tadpole) (**f,l,r**), following ISH with probes for orthologs of catshark markers for Left-LHb (**b–f**), MHb (**h–l**) and Right-LHb (**n–r**). Dorsal is to the top in all panels and probe identity is indicated on each section. Black and white arrowheads in (**b–f**) and (**n–r**) respectively point to territories restricted to the left side and to the right side in the elephant shark, the reedfish and the lungfish, similar to the catshark, but bilateral in the spotted gar and the frog. Black arrows indicate the boundary between dorsal/medial and ventral/lateral habenulae. Dotted lines in (**f,r**) delineate the boundary between the lateral *Sox1* habenular territory and the adjacent *Prox1* territory in the frog. **s–w** Schemes showing territories related to catshark Left-LHb (yellow), MHb (purple) and Right-LHb (blue) in the elephant shark (**s**), the reedfish (**t**), the spotted gar (**u**), the lungfish (**v**) and the frog (**w**), based on the set of signature markers analyzed. Only markers supporting these relationships are indicated for each species, with those also expressed in additional territories labeled by an asterisk (see expression details in Supplementary Fig. 7–11). The same ISH profiles were consistently obtained for each gene and each species on at least three different specimens. Abbreviations: L, left; R, right. Scale bar = 500 μm in (**b,h,n**), 200 μm in (**c,d,i,j,o,p**), 150 μm in (**e,k,q**), 100 μm in (**f,l,r**).

antagonist SB-505124 leads to an expansion of the signal in the Left-LHb, ultimately resulting in a right isomerism (Fig. 3h1–h4). Taken together, these data highlight an asymmetric, dynamic canonical Wnt activity in developing habenulae, and suggest that the marked restriction to the lateral right habenula observed starting from stage 29 is dependent on the transient left Nodal activity, previously reported in the catshark diencephalon at earlier stages of development[37].

### Wnt promotes right identity in catshark lateral habenulae

To test the involvement of right-restricted Wnt activity in the lateral habenula in the elaboration of neuronal identities, we next inhibited the pathway using *in ovo* injection of the Wnt antagonist IWR-1, a tankyrase inhibitor that stabilizes the β-catenin degradation complex via Axin modification[50], and analyzed the resulting phenotype at stage 31 (Fig.4, Supplementary Fig. 13, Supplementary Table 2). A single injection of the drug at stage 29 results in a marked reduction of nuclear β-catenin signal from areas of varying size in the Right-LHb

(compare Fig.4a,a2 and Fig.4h,h2; Supplementary Fig. 13a,e). Expression of the two Left-LHb markers *ScSox1* and *ScNtng2* expands to the right in all embryos injected with the drug (*n* = 9/9; Supplementary Table 2), a phenotype never observed in control embryos (*n* = 8/8; compare Fig. 4b,d,f and Fig. 4i,k,m; Supplementary Fig.13b,d1,f,h1). This expansion of marker gene expression is superimposable on the area devoid of nuclear β-catenin in the Right-LHb (Fig. 4h,i). Partial loss of two Right-LHb markers (*ScProx1* and *ScKiss1*) are consistently observable in the same subdomain (compare Fig. 4c,e,g and Fig. 4j,l,n; Fig. 4k,m and Fig. 4l,n; Supplementary Fig.13c,g).

In the mouse, no asymmetric habenular expression of *Prox1* has been reported, but expression of the gene in adjacent thalamic territories has been shown to be dependent on *Tcf7l2*, in line with a dependence on Wnt signaling[51]. Another gene expressed in a neighboring region of the mouse thalamus, *Rorα*, is submitted to the same regulation. In view of the similarities between the gene signature similarities of the catshark lateral habenula and the mouse thalamic

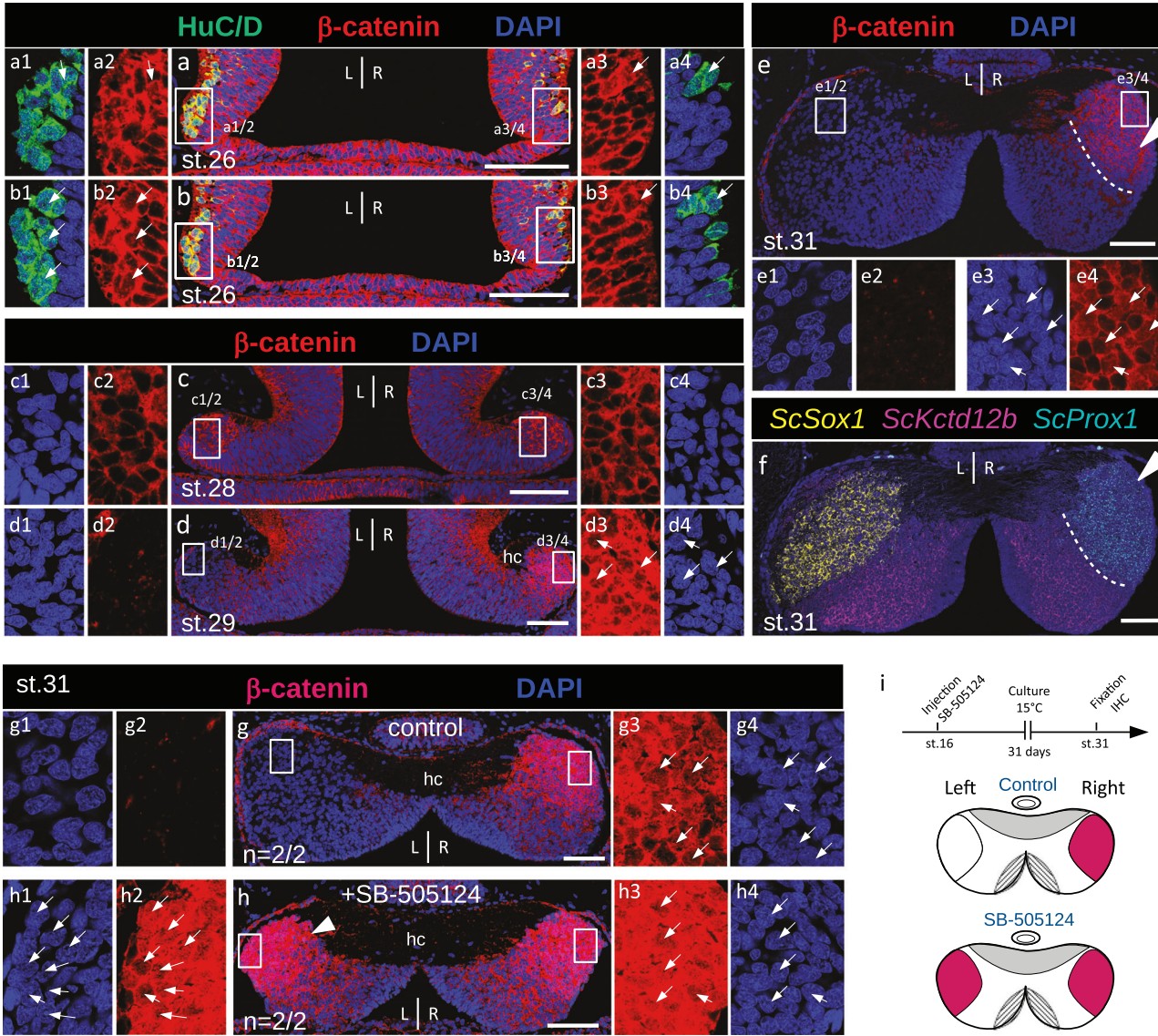

**Fig. 3 | The nuclear β-catenin profile in developing catshark habenulae reveals dynamic, Nodal-dependent asymmetries. a–e** Confocal images of transverse sections of catshark habenulae at stages 26 (**a,b**), 28 (**c**), 29 (**d**) and 31 (**e**) following DAPI staining (blue) and immunohistochemistry (IHC) with antibodies directed against β-catenin (red) (**a–e**) and HuC/D (green) (**a,b**). **f** Hybridization chain reaction-based fluorescent in situ hybridization (HCR-FISH) image of a section adjacent to (**e**) with territories for *ScSox1*, *ScKctd12b*, and *ScProx1* respectively in yellow, magenta and cyan. White arrowheads in (e,f) point to the Right-LHb. Dotted lines in (e,f) delimit Right-LHb and MHb subdomains, respectively positive for *ScProx1* and *ScKctd12b*. The same β-catenin profiles were consistently obtained at each stage on at least three different specimens. **g,h** Confocal images of transverse sections of stage 31 catshark habenulae, after *in ovo* injection of DMSO (**g**, control) or SB-505124 (**h**) following neural tube closure. The section shown in (**a**) is located anteriorly to the one shown in (**b**), sections in (**c–h**) are located at a medial organ level. A white arrowhead in (**h**) points to a lateral left territory showing a high density of β-catenin-positive nuclei, present in SB-505124-treated embryos but absent from control ones. **i** Schemes showing the experimental procedure and the habenular phenotypes observed, with lateral territories of nuclear β-catenin accumulation in red. Dorsal is to the top in all panels. Magnifications of boxed areas in (**a–e,g,h**) are respectively shown in (**a1–a4,b1–b4,c1–c4,d1–d4,e1–e4,g1–g4,h1–h4**). Thin white arrows point to β-catenin-labeled nuclei. Abbreviations: L, left; R, right; st., stage. Scale bar = 100 μm.

regions adjacent to the habenulae, we analyzed expression of the catshark *Rorα* ortholog in control and IWR-1-treated embryos (Supplementary Fig. 13d,h). A lateral right-restricted expression territory of the gene, forming a radial band adjacent to the external part of the MHb, is observed in habenulae of untreated embryos at stage 31 (Supplementary Fig. 2f). This territory, also present in control embryos (Supplementary Fig. 13d2), is reduced following IWR-1 treatment, concomitantly with an expansion of *ScSox1* and a loss of *ScKiss1* expressions in the Right-LHb (compare Supplementary Fig. 13d,h). These data indicate that Wnt signaling is required for the elaboration of right neuronal identities by repressing left ones in the catshark Right-LHb (Fig. 4o). In contrast, we never observed changes of *ScEnpp2*

and *ScPde1a* expressions in the MHb following IWR-1 treatments in these conditions (Supplementary Fig.14).

## Nodal-dependent Wnt repression promotes left identity in catshark lateral habenulae

We previously showed that an early abrogation of Nodal/Activin signaling using *in ovo* injection of the inhibitor SB-505124 during the diencephalic window of left-restricted Nodal activity induces a right isomerism habenular phenotype[37]. To test whether Wnt inhibition could rescue lateral left (Left-LHb) neuronal identities in these embryos, we carried out double *in ovo* injections of SB-505124 (at stage 16) and IWR-1 (at stage 29), and analyzed the resulting phenotype at stage 31 (Fig.5;

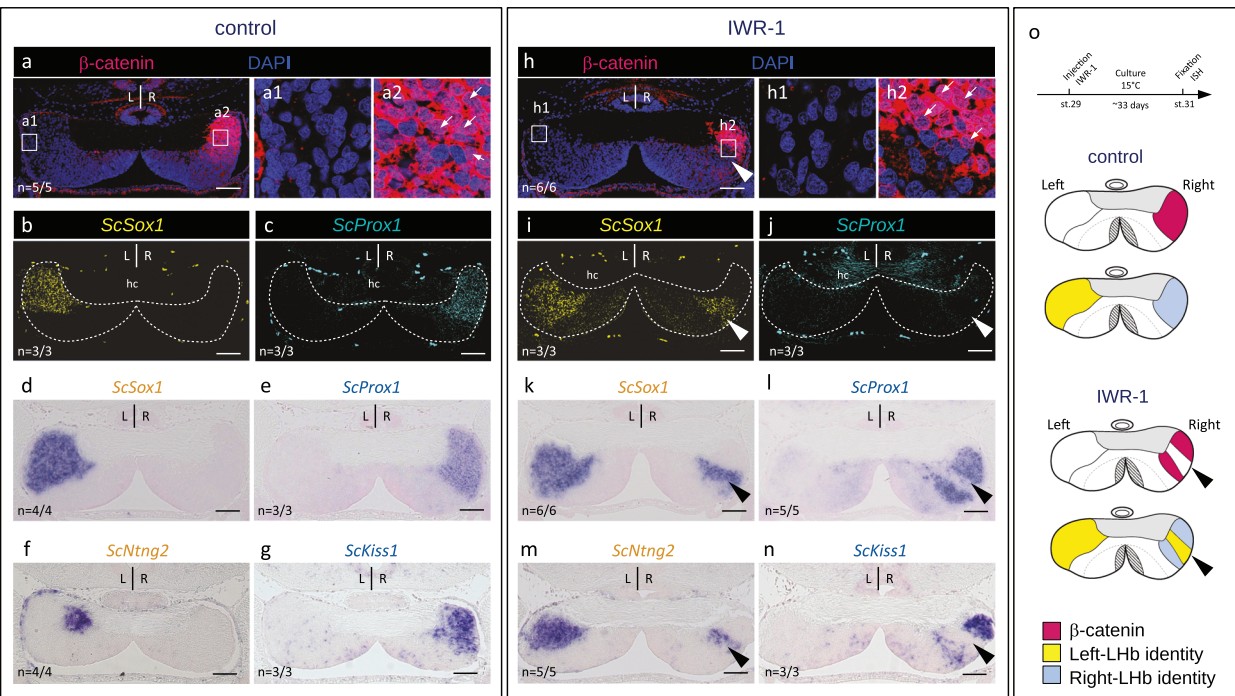

**Fig. 4 | Inhibition of Wnt signaling converts lateral right into lateral left neuronal identities in developing catshark habenulae. a–n** Transverse sections of catshark stage 31 habenulae from control (**a–g**) and IWR-1-treated (**h–n**) embryos, after IHC with an antibody directed against β-catenin in red and DAPI-staining in blue (**a,h**), HCR-FISH double-labeling with territories of *ScSox1* in yellow (**b,i**) and those of *ScProx1* in cyan (c,j), ISH with probes for *ScSox1* (**d,k**), *ScProx1* (**e,l**), *ScNtng2* (**f,m**) and *ScKiss1* (**g,n**). Sections are shown at a medial organ level, dorsal to the top. (**a**) and (**b,c**) show adjacent sections of the same embryo, same for (**d**) and (**e**), (**f**) and (**g**), (**h**) and (**i,j**), (**k**) and (**l**), (**m**) and (**n**). (**a1,a2**) and (**h1,h2**) show magnifications of the territories boxed in (**a**) and (**h**), with β-catenin signals in red and DAPI signals in blue. Thin white arrows point to β-catenin labeled nuclei. Arrowheads in (**i–n**) show expansions of Left-LHb markers to the Right-LHb, in territories where expression of Right-LHb markers and nuclear β-catenin accumulation are lost. "*n* = " in (**a–n**) refers to the number of embryos exhibiting the same phenotype as shown in the figure, over the total number of embryos analyzed, using the same detection method. **o** Schemes showing the experimental procedure used for pharmacological treatments and the resulting habenular phenotypes, with territories of nuclear β-catenin accumulation in red, territories of Left- and Right-LHb identity in yellow and blue respectively. Abbreviations: LHb, lateral habenula; L, left; R, right; st., stage. Scale bar = 100 μm.

Supplementary Fig. 15; Supplementary Table 3). As expected, *in ovo* injections of SB-505124 consistently results in a right isomerism (*n* = 6/6), with a symmetric nuclear distribution of β-catenin in both left and right lateral habenulae (Fig.5a; Supplementary Fig. 15a), a complete loss of lateral left expression of the two Left-LHb markers *ScSox1* and *ScNtng2* (Fig.5b,d,f; Supplementary Fig. 15b,c) and a concomitant lateral left expansion of the Right-LHb markers *ScProx1* and *ScKiss1* (Fig.5c,e,g; Supplementary Fig. 15d). Following IWR-1 injection at stage 29 of SB-505124-treated embryos (*n* = 10/10), habenular symmetry is consistently maintained, but lateral habenulae exhibit non-overlapping territories of Right- and Left-LHb identities, similar to the right habenulae of IWR-1-treated embryos (Fig.5i–n; Supplementary Fig. 15e–h). Compared to SB-505124-treated embryos, nuclear β-catenin becomes undetectable in lateral zones of variable size in both left and right habenulae (compare Fig.5h and 5a; Supplementary Fig. 15a,e). Expression of the Left-LHb markers *ScSox1* and *ScNtng2* is restored in the corresponding territories (compare Fig.5b,d,f and Fig.5i,k,m; Supplementary Fig. 15b,c,f,g), while expression of the Right-LHb markers *ScProx1* and *ScKiss1* is lost (compare Fig.5c,e,g and Fig.5j,l,n; Supplementary Fig. 15b,d,f,h). Taken together, these data suggest that a repression of Wnt signaling is sufficient to promote a lateral left identity in the absence of Nodal signaling (Fig.5o).

### Asymmetric timing of cell cycles exits in catshark medial habenulae

In the zebrafish, Wnt signaling has been shown to delay neuronal differentiation in dorsal habenulae, thus contributing to neuronal diversity and asymmetry[32]. In order to address whether the respective timing of progenitor cell cycle exits differs between catshark left and right habenulae, we conducted BrdU (5-bromo-2′-deoxyuridine) pulse

chase assays starting from early stages of neurogenesis[39] (Fig.6; Supplementary Fig. 16). BrdU incorporation pulses were carried out at stages 26–27 (two stages difficult to distinguish in live embryos), 28 and 29, and the distribution of BrdU-labeled cells was analyzed at stage 31, on transverse sections at different habenula levels. Assignment of labeled cells to the main habenular subdomains was achieved by comparisons to adjacent sections submitted to ISH (Fig. 6).

On the right side, the maximal intensity of BrdU signals is observed along radial bands, which gradually regress from lateral to medial habenula levels at all stages analyzed (Fig. 6a2–a8,b2–b8,c2–c8). A negative territory restricted to posterior and lateral-most Right-LHb territories (*ScProx1*-positive and *ScKctd12b*-negative) is already present following a pulse at stages 26–27 (Fig. 6a4–a8), it expands after a pulse at stages 28 (compare Fig. 6a1–a8 and Fig. 6b1–b8), and the BrdU signal is completely lost in the Right-LHb, becoming restricted to MHb, following a pulse at stage 29 (Fig. 6c1–c8).

On the left side, a similar pattern is observed, with the presence of lateral posterior BrdU-negative territories after pulses at stages 26–27 (Fig. 6a7–a9) and a gradual withdrawal of BrdU-labeled cells from the Left-LHb (*ScSox1*-positive and *ScKctd12b*-positive) after pulses between stages 28 and 29 (compare Fig. 6b1–b9 with Fig. 6c1–c9). However, a major difference is that on the left side, BrdU-negative territories span the anterior-most part of both the lateral (*ScSox1*-positive) and the medial (*ScKctd12b*-positive) habenulae following pulses at all stages analyzed, including stages 26–28 (Fig. 6a1–a6,b1–b6). Furthermore, the sharp boundary between BrdU-positive and -negative cells, observed on the right at all pulse stages and all habenula levels, is absent on the left at anterior to medial organ levels for pulse stages preceding stage 28 (Fig. 6a2,a5).

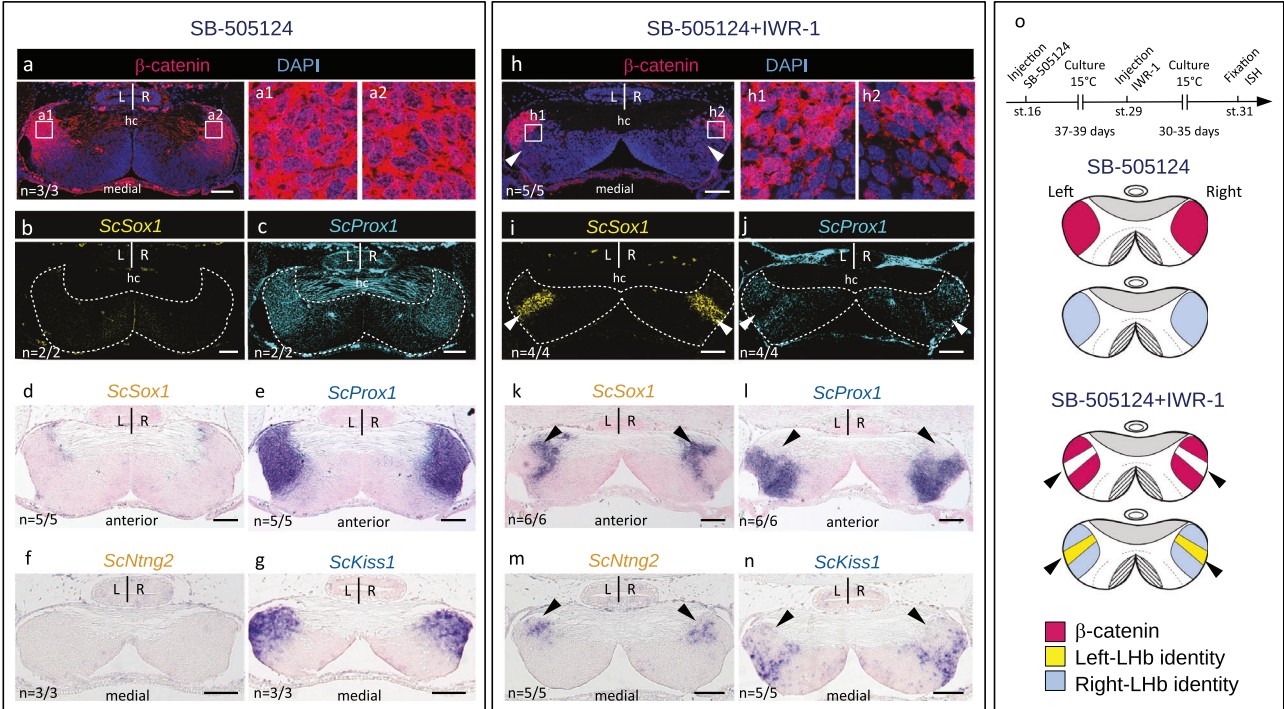

**Fig. 5 | Inhibition of Wnt signaling rescues Left-LHb neuronal identities in the lateral habenulae of SB-505124-treated catshark embryos. a–n** Transverse sections of catshark stage 31 habenulae from SB-505124- (**a–g**) and SB-505124- + IWR-1-treated (**h–n**) embryos, after IHC with an antibody directed against β-catenin (β-catenin in red, DAPI-stained nuclei in blue) (**a,h**), HCR-FISH with territories of *ScSox1* (**b,i**) and *ScProx1* (**c,j**) in respectively yellow and cyan, ISH with probes for *ScSox1* (**d,k**), *ScProx1* (**e,l**), *ScNtng2* (**f,m**) and *ScKiss1* (**g,n**). Sections are shown at anterior (**d,e,k,l**) or medial (**a–c,f–j,m,n**) organ levels, dorsal to the top. (**a**) and (**b,c**) show sections of the same embryo, same for (**d**) and (**e**), (**f**) and (**g**), (**h**) and (**i–j**), (**k**) and (**l**), (**m**) and (**n**). (**a1,a2**) and (**h1,h2**) show magnifications of the territories boxed in (**a**) and (**h**), with β-catenin signals in red and DAPI signals in blue. A right

isomerism is observed following SB-505124 treatment (**a–g**). Arrowheads in (**h–n**) point to bilateral LHb sub-territories showing a Left-LHb identity in embryos co-treated with SB-505124 and IWR-1. "**n** = " in (**a–n**) refers to the number of embryos exhibiting the same phenotype as shown in the figure, over the total number of embryos analyzed, using the same detection method. **o** Schemes showing the experimental procedure used for pharmacological treatments and the resulting habenular phenotypes, with territories of nuclear β-catenin accumulation in red, territories of Left- and Right-LHb identity in yellow and blue respectively. Arrowheads point to bilateral LHb sub-territories in embryos co-treated with SB-505124 and IWR-1. Abbreviations: hc, habenular commissure; L, left; R, right; st., stage. Scale bar = 100 μm.

Analyzes of horizontal sections of stage 31 embryos submitted to BrdU pulses between stages 28 and 29 confirm these conclusions (Supplementary Fig. 16a,b,e,f). BrdU-labeled cells are lost in both the (*ScKctd12b*-negative) Left- and Right-LHb following pulses between stages 28 and 29, regressing from external to internal MHb levels (Supplementary Fig. 16c–f). These data indicate that at anterior-most organ levels, progenitors of both lateral and medial territories exit cell cycles earlier on the left side than on the right side (Fig.6d,e).

**Congruence between Wnt activity and asymmetry pattern in gnathostomes**
In the catshark, IHC profiles of β-catenin remain highly asymmetric in the habenulae of juveniles (Supplementary Fig. 17), with nuclear signals restricted to the lateral right habenula (Supplementary Fig. 17a7,a8,b3,b4,c5,c6). As at stage 31, heterogeneities are also observed in the medial habenula, with cytoplasmic signals being selectively absent in its external anterior left component (Supplementary Fig. 17a,a3,a4). In order to test whether the presence of asymmetries in ventral/lateral habenulae correlates with an asymmetric distribution of β-catenin across gnathostomes, we performed IHC analyzes on transverse sections from reedfish and spotted gar juveniles, as well as frog tadpoles (stage NF-66) (Fig.7). The lungfish was excluded from this analysis as no positive control signal could be obtained in this species.

In the reedfish, no nuclear signal could be observed on the left (Fig.7a1,a2). In contrast, most of the right-restricted ventral territory of *Kiss1* expression contains a high density of β-catenin-positive nuclei

(Fig.7a,a3,a4,7b;). In the dorsal right *EcKiss1*-negative territory, nuclei are generally unlabeled except in a limited lateral area (Fig.7a5,a6). In contrast, in the spotted gar, nuclear distribution of β-catenin is observed both on the right and on the left, in ventral *Kiss1*-positive territories (Fig.7c1–c4,7d). As in the reedfish, dorsal *LoKiss1*-negative territories are heterogeneous, with some areas, such as the dorsal-most part of the left dorsal habenula, devoid of signal (Fig.7c), and others containing either cytoplasm-restricted or nuclear β-catenin signals (Fig.7c5,c6). In the frog, no nuclear signals were observed in the dorsal-most *Kctd*-positive or in the adjacent *Sox1*-positive domains (Fig.5e,e1,e2,f,f1,f2,f3,f4). However, a significant proportion of β-catenin-positive nuclei is present in adjacent ventral territories, corresponding to those expressing *Prox1* (Fig.7f5,f6,f7,f8). In conclusion, right-restricted nuclear accumulations of β-catenin, maintained until advanced stages of differentiation, correlate with the presence of asymmetric neuronal identities in ventral/lateral habenulae.

**Partial conservation of habenular asymmetries in cyclostomes**
To further explore the origin of the molecular asymmetries shared by the catshark, the elephant shark, the lungfish and the reedfish in ventral/lateral habenulae, we assessed their conservation in a cyclostome, the river lamprey *Lampetra fluviatilis*. We first conducted phylogenetic analyzes to determine the full repertoire of lamprey genes in the *Pcdh10/10 l/17/18/19*, *Sox1/2/3*, *Ntng1/2*, *Prox1/2* and *Kctd8/12/12b/16* gene families (Supplementary Fig. 18) and surveyed expression of each paralog in adult habenulae, as well as in the pineal and parapineal organs, as assessed from single-nuclei RNA-seq data available for the

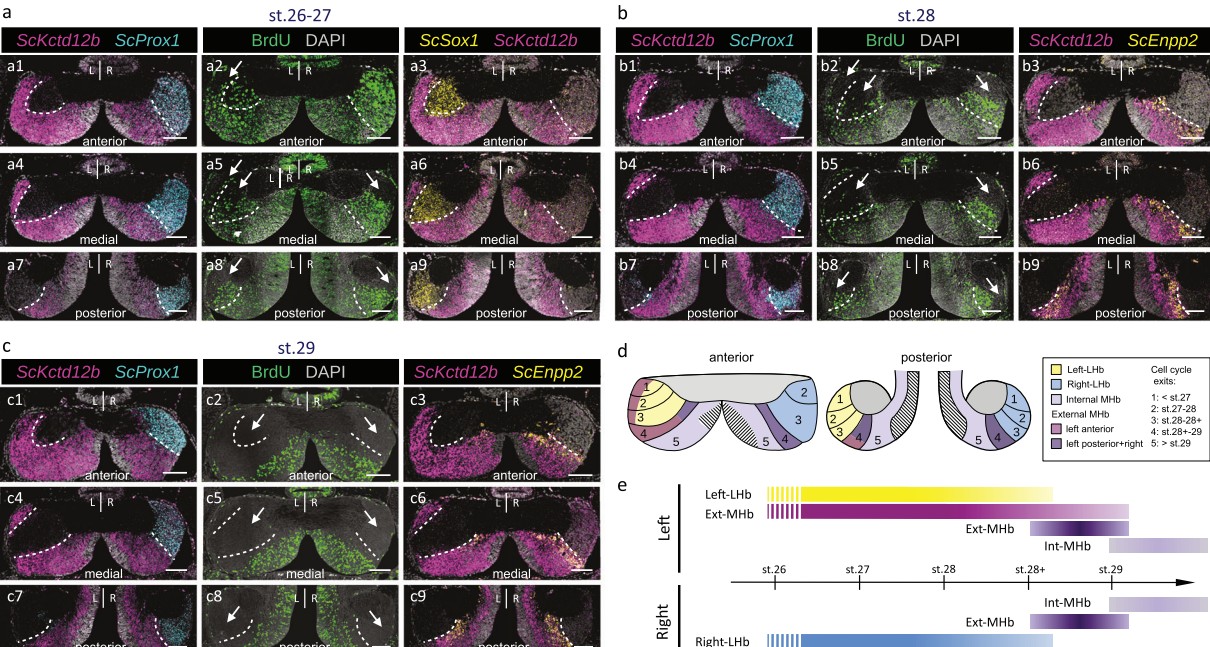

**Fig. 6 | Spatial and temporal regulation of progenitor cell cycle exits in developing catshark habenulae. a–c** Transverse sections of catshark stage 31 habenulae following exposure of embryos to BrdU pulses at stage 26−27 (**a**), 28 (**b**), and 29 (**c**), dorsal to the top. (**a2,a5,a8,b2,b5,b8,c2,c5,c8**) show confocal images following IHC using an antibody directed against BrdU (green), with DAPI-stained nuclei shown in gray. (**a1,a4,a7,b1,b4,b7,c1,c4,c7**), (**a3,a6,a9**), and (**b3,b6,b9,c3,c6,c9**) respectively show confocal images after double fluorescent ISH with probes for *ScKctd12b/ScProx1* (magenta/cyan), *ScSox1/ScKctd12b* (yellow/ magenta), and *ScKctd12b/ScEnpp2* (magenta/yellow), with DAPI-stained nuclei in gray. (**a1,a2,a3**) show adjacent sections at an anterior level, same for (**b1,b2,b3**) and (**c1,c2,c3**). (**a4,a5,a6**) show adjacent sections at a medial level, same for (**b4,b5,b6**) and (**c4,c5,c6**). (**a7,a8,a9**) show adjacent sections at a posterior level, same for (**b7,b8,b9**) and (**c7,c8,c9**). White dotted lines delimit the border between medial (MHb) and lateral (LHb) territories as inferred from *ScKctd12b* expression, and its approximate location in adjacent BrdU-labeled sections. Thin arrows point to BrdU-negative territories. **d** Scheme showing the spatial distribution of territories exiting cell cycles earlier than stage 27 (1), during stages 27-28 (2), stages 28-28+ (3), stages 28 +-29 (4) and later than stage 29 (5), superimposed on the subdomain organization of catshark stage 31 habenulae. **e** Scheme showing the developmental windows when the broad territories of stage 31 catshark habenulae exit cell cycles. Data from Supplementary Fig. 16 are taken into account in (**d**) and (**e**). Color code in (**d,e**) as in Fig. 1a2,b2. Assays at each incorporation stage were replicated at least three times, with the same results. Abbreviations: LHb, lateral habenula; MHb, medial habenula; L, left; R, right; st., stage. Scale bar = 100 μm.

lamprey brain[52] (Supplementary Table 4). We then selected one to three lamprey paralogs within each family for ISH analysis, always including those paralogs with expression in habenula cell clusters. Two putative lamprey homologs of gnathostome *Kiss1/2*[53] were also added to this ISH analysis.

Selective expression in parapineal or pineal cell clusters is documented for at least one gene belonging to the *Sox1/2/3/19* and *Ntng1/2* families, but members of the *Pcdh10/10 l/17/18/19*, *Sox1/2/3/19* and *Ntng1/2* families show no sustained expression in habenular cell clusters (proportion of expressing cells>0.5) except for two genes, respectively annotated as *Pcdh17l* and *Ntng2l*, both of which are broadly expressed in diencephalic cell clusters (Supplementary Table 4). ISH of transverse sections of lamprey adult habenulae confirmed these results, with either a complete absence of expression, or conspicuous habenular expression for *LfPcdh17l* and *LfNtng2l*. The ISH survey of the lamprey *Kiss1/2* homologs also yielded no detectable expression.

In contrast, strongly asymmetric expression was obtained for the *Prox1/2* and *Kctd8/12/12b/16* family members tested, respectively referred to as *LfProx1la*, *LfProx1lb* and *LfKctd12l* (Supplementary Table 4; Fig.3). *LfProx1la* and *LfProx1lb* territories are restricted to the right habenula, albeit with distinct, overlapping profiles (Fig.8c,d). *LfProx1lb* is broadly expressed in the right habenula, only excluding a medial dispersed cell population positive for *LfKctd12l* in a tract-rich zone (Fig.8a2,b2,d2). The *LfProx1la* expression territory is very similar anteriorly (Fig. 8c1), but restricted to two discrete zones posteriorly, a lateral one and a more medial one, also excluding the medial dispersed cell population devoid of *LfProx1lb* expression (Fig. 8c2,c3). *LfKctd12l*

expression spans the complete left habenula (Fig.8b). In addition, on the right side, the gene shows a faint, broadly distributed signal, with a total absence in a lateral *LfProx1la*-positive territory (Fig.8b2,c2) and a higher intensity in three distinct zones: the medial dispersed cell population excluding *LfProx1la* and *LfProx1lb* expressions (Fig.8a2,b2), and two zones positive for *LfProx1lb* but not for *LfProx1la*, an anterior ventral one (Fig.8a1,b1,d1) and a posterior one adjacent to the midline (Fig.8a2,a2′,a3,b2,b3,d2,d3).

We further used IHC to examine the nuclear distribution of β-catenin in adult habenulae of the lamprey (Fig.8e). As in the catshark and the reedfish, β-catenin nuclear signals are undetectable in the left habenula (Fig.8e1−e3,e1′). In the right habenula, nuclear signals are widespread, but the proportion of positive nuclei varies depending on the territory considered (compare Fig.8e1″,e2′, e3′). For instance, we could not detect β-catenin-expressing nuclei in the medial *LfKctd12l*-positive/*LfProx1la/b*-negative cell population (Fig.8e2), nor in the posterior *LfKctd12l/LfProx1lb* territory (Fig.8e3,e3′), while about 80% of the nuclei exhibit strong β-catenin signals in the lateral *LfProx1la* territory negative for *Kctd12l* (Fig.8e2,e2′). In summary, our data suggest that right-restricted neuronal identities and Wnt activity may be shared between the lamprey and gnathostomes, despite a very different relative organization of habenular subdomains (Fig.8f).

## Discussion
We show that catshark habenulae exhibit a highly asymmetric organization, with many of the genes identified here as asymmetrically expressed in this species being reported as such for the first time in a vertebrate. Asymmetric profiles concern not only the medial (dorsal in

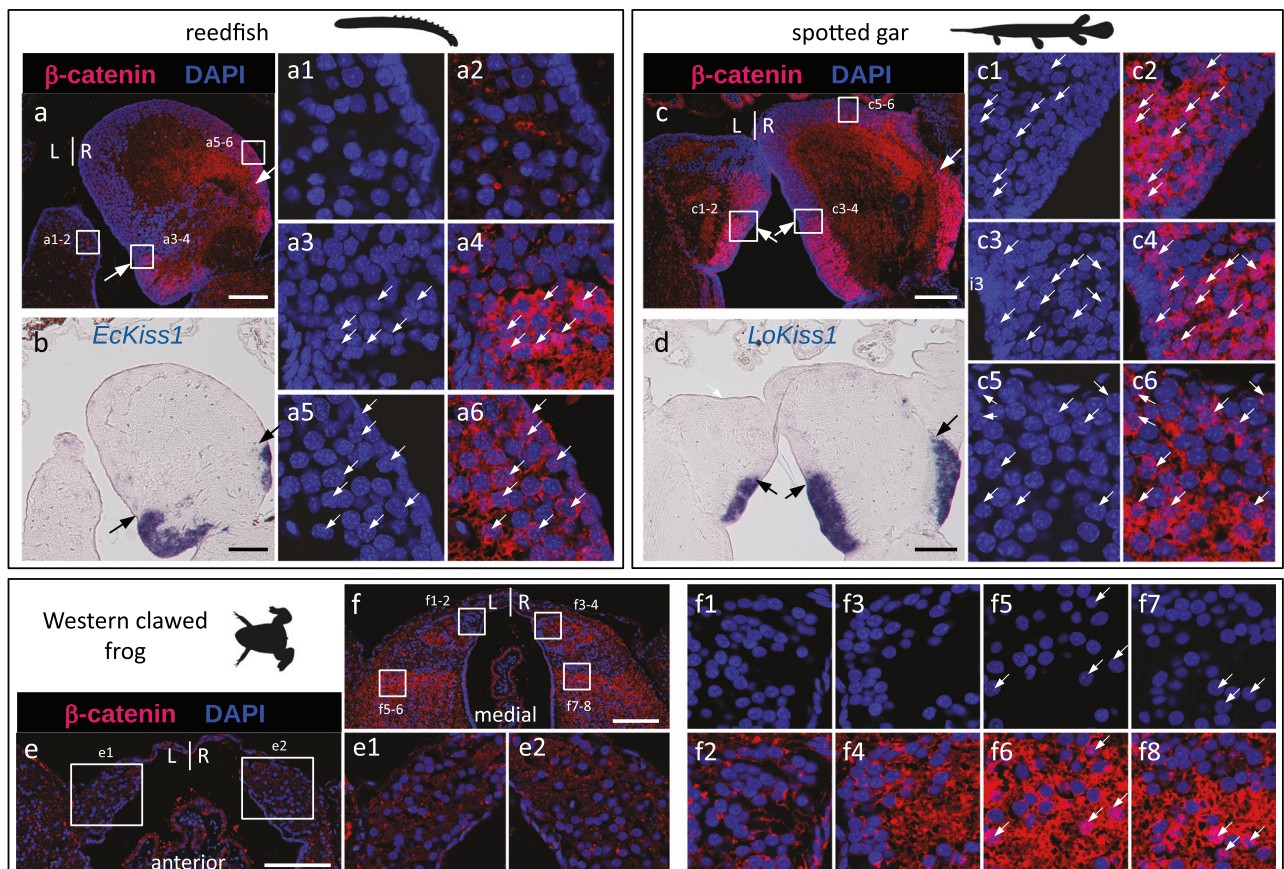

**Fig. 7 | Evolution of nuclear β-catenin asymmetry patterns in the habenulae of jawed vertebrates. a–f** Transverse sections of habenulae from the reedfish (juvenile) (**a**,**b**), the spotted gar (juvenile) (**c**,**d**) and the Western clawed frog (NF-66 tadpole) (**e**,**f**), after IHC using an antibody directed against β-catenin (red) with DAPI-stained nuclei in blue (**a**,**c**,**e**,**f**) and after ISH with probes for *Kiss1* orthologs (**b**,**d**). Dorsal is to the top in all panels. Arrows in (**a**,**b**,**c**,**d**) indicate the boundary between ventral, *Kiss1*-positive territories, and dorsal territories. (**a1–a6**), (**c1–c6**), (**e1–e2**) and (**f1–f8**) show higher magnifications of territories boxed in (**a**), (**c**), (**e**) and (**f**) respectively, with DAPI staining shown in (**a1–a3,c1,c3,c5,f1,f3,f5,f7**) and merged signals for DAPI (blue) and β-catenin (red) shown in (**a2,a4,a6,c2,c4,-c6,e1,e2,f2,f4,f6,f8**). White arrows in (**a3–a6,c1–c6,f5–f8**) point to β-catenin-positive nuclei. Identical nuclear β-catenin profiles were consistently obtained for each species on at least two specimens. Abbreviations: L, left; R, right. Scale bars = 100 μm.

teleosts) habenulae as in the zebrafish[19,22,24,25], but also the lateral habenulae, which in the catshark, exhibit distinct neuronal identities between the left and right sides, contrary to their counterpart (the ventral habenulae) in teleosts. Analysis of a wider range of gnathostomes provides a broad evolutionary overview, shedding light on the mode of evolution underlying this divergence between the zebrafish and the catshark. In the lateral habenulae, asymmetries related to those observed in the catshark are detected, with the same laterality, not only in the elephant shark, a member of the holocephalans (which diverged from elasmobranchs about 410 Mya[54]), but also in a sarcopterygian, the African lungfish, and in an actinopterygian, the reedfish. This similarity contrasts with the bilateral patterns observed in lateral habenulae and adjacent ventral territories of the frog and the mouse, or in the ventral habenulae of the spotted gar and teleosts, respectively more similar to the catshark left or right side.

We cannot rule out convergent evolutionary processes to explain the similarities of asymmetric patterns in the lateral habenulae of chondrichthyans, lungfishes and polypterids. However, the phylogenetic distribution of the conservations, shared by chondrichthyans and early diverging lineages in actinopterygians and sarcopterygians, does not evoke a punctuated presence/absence pattern, as observed for instance for the left-restricted nucleus recently identified in teleosts[24]. Furthermore, the homogeneity in the relative organization of habenular subdomains observed in the mouse and the frog, on the one hand, and in neopterygians, including several teleosts and the spotted

gar, on the other hand[24], argues against a rapid drift of the broad architecture of habenulae in these taxa.

Comparisons at a wider evolutionary scale, between the catshark and the river lamprey, also reveal similarities in asymmetry patterns, with the restriction to the right side of a broad territory related to the catshark lateral right habenula and a bilateral distribution of territories expressing members of the *Kctd8/12/12b/16* family. However, we found no evidence for habenular territories related to the catshark left lateral habenula, with the whole lamprey left habenula expressing *LfKctd12l*. Similarly, contrary to gnathostomes, we observed several nuclei of mixed identity, co-expressing *LfKctd12l* and a *Prox1* family member. The organization of the lamprey habenulae emerging from these molecular comparisons, although based only on a limited number of markers, is consistent with a projection analysis, which showed a right restriction of territories related to the gnathostome lateral habenula and a circuitry reminiscent of those involving the gnathostome medial habenula on the left[29,30].

Taken together, these data suggest a multi-step mode of habenular asymmetry evolution involving (1) an ancient restriction in vertebrates of neuronal lateral right identities to the right side in habenulae already endowed, as previously proposed[29], with a bipartite organization, (2) the fixation, early in the gnathostome lineage, of bilateral medial and lateral subdomains, in the relative topological organization observed in all extant gnathostomes, and of lateral habenula asymmetries related to those reported in the catshark, (3) the

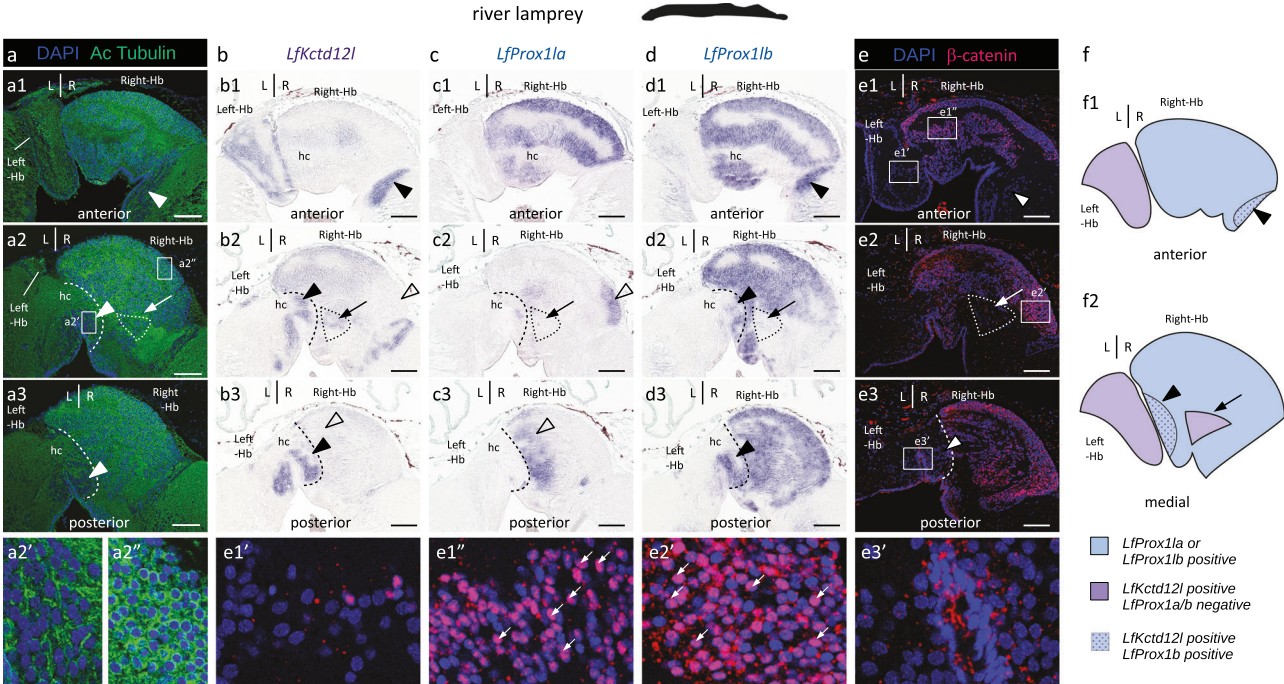

**Fig. 8 | Expressions of *Prox1* paralogues and nuclear distribution of β-catenin are right-restricted in the habenulae of the river lamprey. a,e** Confocal images of transverse sections of adult river lamprey habenulae following DAPI staining (blue) and IHC with antibodies directed against acetylated tubulin (green) (**a**) and β-catenin (red) (**e**). Higher magnification of the territories boxed in (**a2**), (**e1**), (**e2**) and (**e3**) are shown in (**a2'**,**a2"**), (**e1'**,**e1"**), (**e2'**) and (**e3'**), respectively. **b,c,d** Transverse sections of river lamprey habenulae following ISH with probes for *LfKctd12l* (**b**), *LfProx1la* (**c**) and *LfProx1lb* (**d**). Sections in (**b,c,d**) were obtained from the same specimen. Sections shown in (**a1–a3**), (**b1–b3**,) (**c1–c3**), (**d1–d3**), and (**e1–e3**) are anterior to posterior series. White arrrowheads in (**a**) and black arrowheads in (**b,d**) point to two territories, which express both *LfKctd12l* and *LfProx1lb* in the right habenula: one, delimited by dashed lines in (**a2,a3,b2,b3,c2,c3,d2,d3,e3**), is located posteriorly and adjacent to the midline, while the other, shown in (**a1,b1,d1,e1**), is restricted to ventral and anterior levels. Empty arrowheads in (**b2,b3,c2,c3**) point to discrete posterior *LfProx1la* territories negative for *LfKctd12l*. A thin arrow in (**a2,b2,c2,d2,e2**) points to a right medial territory, delimited by a dotted line, which expresses *LfKctd12l*, but neither *LfProx1la*, nor *LfProx1lb*. Small arrowheads in (**e1',e2'**) point to β-catenin- positive nuclei. **f** Schemes showing the subdomain organization of river lamprey habenulae observed on transverse sections at anterior and medial levels. Color code: blue, territories expressing at least one of the two *LfProx1la*/b paralogs; purple, territories expressing *LfKctd12l*, but neither *LfProx1la*, nor *LfProx1lb*; dotted purple/blue, territories expressing both *LfKctd12l* and *LfProx1lb*. Each ISH and IHC experiment was replicated on at least three different specimens with the same results. Abbreviations: Hb, habenula; hc, habenular commissure; L, left; R, right. Scale bar = 100 μm.

occurrence of deviations from this ancestral gnathostome pattern in tetrapods and neopterygians, with independent losses of lateral habenula asymmetries and distinct evolutionary trajectories in these two phyla (Fig. 9a).

Concerning diversifications observed in lateral habenulae, it is intriguing that in the mouse, several orthologues of catshark Right-LHb markers (*Prox1*, *Rora* and *Rerg*) are not expressed in habenulae but rather in the adjacent paraventricular nucleus of thalamus[51]. We observed the same relative location of the *Prox1* territory, adjacent to *Sox1* expression in lateral habenulae of the frog, although the precise posterior and ventral boundaries of habenulae are difficult to identify due to the absence of marked morphological landmarks. Similarly, in the lamprey, specific expressions of *Sox1/2/3/19* and *Ntng1/2* family members were reported in parapineal/pineal cell clusters, while expression of these genes remained undetectable in habenulae. These observations raise the possibility that ancient regulatory circuits, controlling neuronal identities related to those observed in catshark lateral habenulae, may be deployed in different territories, habenular, pineal/parapineal or thalamic, symmetric or asymmetric, depending on the species.

This evolutionary scenario leaves several questions unanswered. First, the organization and the asymmetries of lamprey habenulae may equally reflect the vertebrate ancestral state, or a derived condition from an ancient gnathostome-like pattern. This point could be clarified by analyzes of hagfishes, the sister group of lampreys in cyclostomes. Second, whether asymmetries in medial habenulae followed the same evolutionary trajectories as in lateral habenulae is uncertain. We found no evidence for a strict conservation of asymmetry lateralities for orthologs of catshark asymmetric MHb markers in the elephant shark, and, even across teleosts, asymmetries in dorsal habenulae appear submitted to important variations[25,55,56]. Similarly, conspicuous asymmetries in cellular organization were reported in the dorsal/medial habenulae of some amphibian species[57,58] as well as in some lizards[59], but without evidence of conservation within these taxa. Finally, there is also no evidence for a relationship between the marked molecular asymmetries reported here in the catshark and the subtle ones recently reported in mammalian habenulae, including their lateral component[15,26,27], which may have completely different evolutionary and developmental origins.

Despite the variations of habenular asymmetries, Wnt signaling may be a recurrent factor involved in their formation. Concerning lateral habenulae, our data demonstrate a role for the Wnt pathway in promoting right versus left neuronal identities in the catshark (Fig. 9b). Contrary to the involvement of Wnt signaling in the elaboration of asymmetries in the zebrafish dorsal habenula[32,36,45], this effect is not related to perturbations of the temporal regulation of neurogenesis, but rather results from an asymmetric choice of neuronal identities in post-mitotic neurons. Accordingly, our BrdU analyzes show synchronous cell cycle exits for left and right lateral habenula progenitors and indicate that right and left lateral habenula precursors are post-mitotic at the stage when Wnt signaling is inactivated.

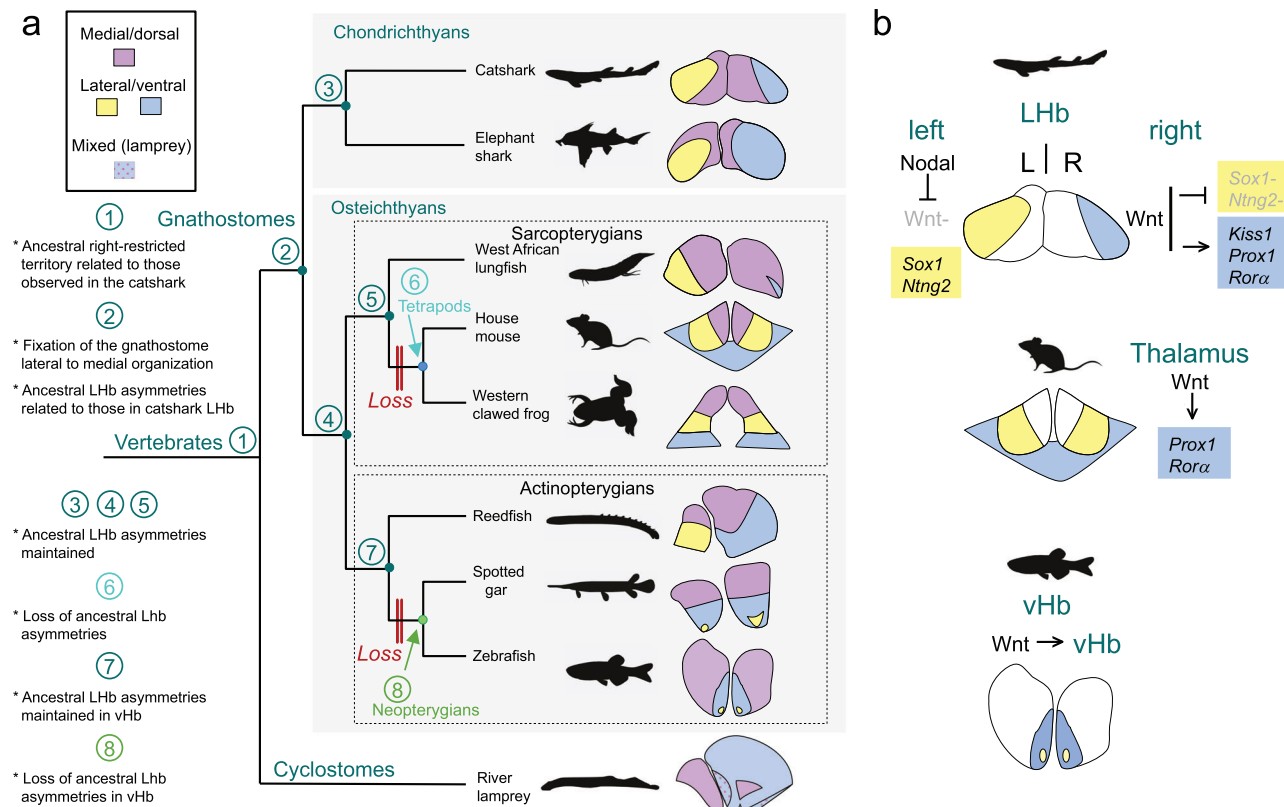

**Fig. 9 | Evolution of habenular asymmetries in jawed vertebrates. a** Evolution of the organization and the asymmetries of habenulae across vertebrates. Schemes on the right show the general subdomain organization of habenulae in members of the major vertebrate phyla. In gnathostomes, medial/dorsal territories are shown in purple, and lateral/ventral territories in yellow for territories of neuronal identity related to catshark Left-LHb or blue for territories of neuronal identity related to catshark Right-LHb. In the lamprey, territories related to the catshark medial and lateral right habenulae (respectively in purple and blue) are shown with the same color codes. Territories of mixed identity, co-expressing markers of medial and lateral right habenulae, are shown dotted in this species. Numbers at the nodes of the tree refer to the ancestral asymmetry profile inferred from comparative analyzes. The phylogenetic distribution of asymmetries suggests that a right restriction of neuronal identities related to those observed in the catshark lateral right habenula is an ancestral vertebrate feature (node 1 of the tree). The typical lateral to medial organization of habenulae, with lateral habenula asymmetries related to

those observed in the catshark, were fixed in the gnathostome lineage, prior to its radiation (node 2). Lateral habenula asymmetries were maintained in chondrichthyans (node 3), as well as in ancestral osteichthyans, actinopterygians and sarcopterygians (nodes 4,5,7). They were independently lost in tetrapods (node 6) and neopterygians (node 8). **b** Differential deployment of an ancestral Wnt dependent regulatory module across jawed vertebrates. Functional analyzes in the mouse and in the catshark suggest that the Wnt signaling dependence of *Prox1/Rorα* expressions may reflect an ancient regulatory module, recruited to shape neuronal identities in the dorsal thalamus of tetrapods. In the catshark, the lateral right *Kiss1* expression is also Wnt-dependent and the repression of Wnt by Nodal on the left side results in lateral *Sox1* and *Ntng2* expressions. In the zebrafish, Wnt signaling is required for ventral habenula formation, and possible later roles of the Wnt pathway in the elaboration of neuronal identities in this territory remain to be assessed. Abbreviations: LHb, lateral habenula; vHb, ventral habenula; L, left; R, right.

Whether a similar Wnt-dependent regulation of neuronal identities in post-mitotic neurons takes place in the ventral habenula of zebrafish, molecularly related to the catshark lateral right habenula, remains to be assessed, since the loss of this territory in *tcf7l2* mutants precludes an easy dissection of the successive functions of this gene in this species[33]. In contrast, a Wnt-dependent regulation reminiscent of the one identified in catshark lateral habenulae has previously been demonstrated in the mouse, where *Tcf7l2* is required for the refinement of neuronal identities and for the regulation of *Prox1* and *Rorα* expression in post-mitotic neurons of thalamic nuclei adjacent to the habenula[51,60].

The conservation across species of nuclear β-catenin signals in territories showing molecular similarities to the catshark lateral right habenula (i.e., in lamprey right habenulae, reedfish right ventral habenulae, spotted gar ventral habenulae, *Prox1*-expressing region adjacent to the frog lateral habenulae) is also consistent with the hypothesis of an ancestral Wnt-dependent genetic program, promoting a right neuronal identity in lateral habenulae, even though the cellular context may vary across species. We therefore propose that the Wnt signaling-dependence of lateral right neuronal identities

observed in the catshark reflects an ancient regulation, which was already involved in the elaboration of related, right-restricted neuronal habenular identities in ancestral vertebrates, and which was subsequently deployed in different, bilateral territories of tetrapods (Fig. 9b).

Concerning asymmetries in the medial/dorsal habenulae, even though their mode of diversification remains unclear, their formation may also involve conserved mechanisms. In this case, however, the underlying cellular mechanism might be an asymmetric temporal control of neurogenesis, as described in the zebrafish[32,36,45], rather than an asymmetric control of cell fate choices in post-mitotic neurons as observed in catshark lateral habenulae. In support of this hypothesis, we observe a strict lateral to medial temporal sequence of neurogenesis in the catshark right habenula, with successive cell cycle exits of progenitors respectively contributing to lateral, external medial and internal medial habenula components. This sequence appears modified on the left side, with simultaneous, early emergence of progenitors of the lateral left habenula and of the anterior, left-restricted external medial territory.

This temporal regulation is reminiscent of the delay of neuronal differentiation reported in the zebrafish right versus left dorsal habenula[45]. Comparisons with mouse habenulae, which similarly involve a (symmetric) sequential differentiation process during development[16], also highlight similarities between the catshark early-emerging, left-restricted medial habenula component and the lateral-most parts of mouse medial habenulae, with shared expressions of *Spon1* and *Trhde* orthologs. Thus, as observed in lateral habenulae, the mouse is more similar to the catshark left side in medial habenulae.

This similarity in lateral to medial organization also supports the existence of an ancient temporal control of neurogenesis, shaping neuronal identities in medial habenulae. Whether this neurogenetic temporal regulation is under the influence of an asymmetric, right-sided Wnt activity, resulting in asymmetric neuronal identities, as demonstrated in the zebrafish, remains to be assessed in the catshark. In the absence of functional information, this hypothesis is nonetheless consistent with the specific absence of cytoplasmic β-catenin in the left-restricted component of the catshark external medial habenula (but not in its right-enriched complement), which evokes a differential regulation of Wnt pathway components in this territory.

In summary, despite an extensive divergence of habenular asymmetries between the zebrafish and the catshark, their mechanisms of formation display remarkable similarities, which likely reflect ancestral traits. Those include an asymmetric temporal regulation of neurogenesis in medial/dorsal habenulae, a highly dynamic regulation of Wnt signaling and a left-restricted repression of the Wnt pathway, resulting in asymmetric modifications of neuronal identities, albeit via different cellular mechanisms in medial and lateral habenular contexts.

However, a major difference is that Wnt signaling is submitted to different asymmetric regulations in the two species, respectively Nodal-dependent in the catshark and parapineal-dependent in the zebrafish[35,37]. This difference is not absolute, as an early neurogenetic asymmetry in the zebrafish is Nodal-dependent[61], possibly reflecting a vestigial trait, inherited from an ancient Nodal-dependent regulation of neurogenesis on the left. In addition, it might be that, similar to the role of *sox1a* in the zebrafish parapineal[62], the left-restricted expressions of *Sox1* in the catshark left lateral habenula and of *Sox1/2/3/19* family members in lamprey parapineal cell clusters[52], mediate a left-sided repression of Wnt activity in these species.

On a morphological level, in a structure forming through sequential neurogenesis and neuronal differentiation processes such as the habenulae, the highly dynamic regulation of Wnt signaling, prone to variations in time and space, and the pleiotropy of its effects on different successive progenitor states, are substrates for diversifications of asymmetries, as suggested in teleosts[56]. Such variations might in turn lay the ground for major mechanistic transitions, such as the switch from a Nodal-dependent left repression of Wnt signaling, as observed in the catshark[37], to a parapineal-dependent repression, as described in the zebrafish[35]. Understanding how such developmental changes, affecting a morphological trait known to regulate important organismal responses to environmental cues, correlate with diversifications of neuronal circuitry and of behavioral adaptations in different ecological contexts, will be an important challenge for the future.

## Methods

### Animal and embryo collection
This work complies with all relevant ethical regulations. *Scyliorhinus canicula* eggs and juveniles were provided by the Aquariology Service of the Banyuls-sur-Mer Oceanological Observatory (agreement number A6601602) and their analysis was conducted in CNRS Sorbonne Université UMR7232-Biologie Intégrative des Organismes Marins. Ethical review and agreement were not required for analyzes of catshark specimens according to French and European regulations because their study only involved analyzes of non-feeding embryos or

tissue (brain) collection from euthanized juveniles. Embryos were staged according to[63]. *Lampetra fluviatilis* adults and juveniles of *Protopterus annectens*, *Lepisosteus oculatus, Erpetoichthys calabaricus* were obtained from commercial sources and their analysis was conducted in CNRS Sorbonne Université UMR7232-Biologie Intégrative des Organismes Marins. Ethical review and agreement were not required for analyzes of these specimens according to French and European regulations because their study only involved analyzes of tissue (brain) collection from euthanized animals. Impregnated *Callorhinchus milii* females were caught by rod and reel from Western Port Bay, Victoria, Australia according to fishing permits DPI RP1000, RP1003, and RP1112 (Department of Primary Industries, Victoria, Australia) and ethics license Permit MAS/ARMI/2010/01 from Monash University Animal Ethics. The eggs were tagged on the day when they were laid and developed in a recirculating, temperature-controlled aquarium at Monash University as previously described[64]. The eggs were opened, the embryos dissected and staged according to[65]. *Xenopus tropicalis* tadpoles were obtained, euthanized and fixed in the University of Concepcion, Chile, according to ethics license permit CEBB-409-2019 from University of Conception Comité de Etica, Bioética y Bioseguridad. All specimens used in this study (embryos, juveniles, adults) were euthanized with an overdose of tricaine.

### RNA isolation
Left and right habenulae were manually dissected from a total of 45 anaesthetized stage 31 catshark embryos and stored in RNAlater (Thermo Fisher Scientific, Carlsbad, CA, USA) until RNA extraction. Three left pools, each containing 15 left habenulae and three right pools, containing the corresponding 15 right habenulae, were prepared from these explants. Total RNA was extracted from these pools using the Ribopure Kit (Thermo Fisher Scientific, Carlsbad, CA, USA). RNA quantities and integrity indexes (RINs) were controlled using a Bioanalyzer 2100 (Agilent Technologies, Santa Clara, CA, USA). RNA quantities ranging between 1.1 and 2.4 µg and RIN values between 7.4 and 9.6 were obtained for the analyzed pools.

### Library construction and sequencing
We prepared RNA-seq libraries using the Truseq Stranded mRNA Sample Prep Kit (Illumina, San Diego, CA, USA) according to the manufacturer's instructions. Briefly, polyadenylated RNAs were selected using oligo-dT magnetic beads. The polyA+ RNAs were fragmented using divalent cations at elevated temperature and reverse transcribed using random hexamers, Super Script II (Thermo Fisher Scientific, Carlsbad, CA, USA) and actinomycin D. Deoxy-TTP was replaced by dUTP during the second strand synthesis to prevent its amplification by PCR. Double stranded cDNAs were adenylated at their 3' ends and ligated to Illumina's adapters containing unique dual indexes (UDI). Ligated cDNAs were PCR amplified in 15 cycles, and the PCR products were purified using AMPure XP Beads (A63880, Beckman Coulter Genomics, Brea, CA, USA). The size distribution of the resulting libraries was monitored using a Fragment Analyzer (Agilent Technologies, Santa Clara, CA, USA) and the libraries were quantified using the KAPA Library Quantification Kit (Roche, Basel, Switzerland). The libraries were denatured with NaOH, neutralized with Tris-HCl, and diluted to 7.5 pM. Clustering was carried out on a cBot and sequencing was performed on a HiSeq 2500 (Illumina, San Diego, CA, USA) using the paired-end 2×125 nt protocol on two lanes of a flow cell (SRA: PRJNA1040760).

### Read mapping and expression profiling
Reads obtained from each one of the three left and right habenula pools were pseudo-mapped onto an annotated database of reference gene models[66], and pseudo-counted using a k-mer quantification method, kallisto[67]. Contigs exhibiting statistically significant count differences between the left and right habenulae were identified using

the Wald test (q-value threshold 5E-02) implemented in sleuth[68]. We referred to these genes as left- or right-enriched, by comparison to the contralateral side.

### Gene ontology (GO) analysis

Left versus right habenulae GO term enrichment gene analysis was carried out using the ConsensusPathDB online tool (http://cpdb. molgen.mpg.de/)[69,70] against the whole list of genes in the catshark gene model reference. It was restricted to levels 2-5 GO terms related to biological processes (p-value cut-off 5E-02, further curated to a q-value threshold of 5E-02).

### In situ hybridization (ISH), hybridization chain-reaction based fluorescent in situ hybridization (HCR-FISH) and immunohistochemistry (IHC)

Probes were obtained from collections of embryonic *S. canicula* cDNA recombinants[71], or obtained from synthetic double-stranded DNA (Supplementary Data 2). Following ISH, nuclei were counterstained using Nuclear Fast Red Solution (Sigma-Aldrich, Saint-Louis, MO, USA). Chromogenic ISHs of paraffin sections were carried out using digoxigenin-labeled antisense RNAs and standard protocols as previously described[37]. Fluorescent ISHs were performed using digoxigenin- and fluorescein-labeled antisense RNAs following a published protocol[72]. HCR-FISH were carried out using published methods[73,74] with slight modifications: slides were incubated for 20 min in 10 µg/mL proteinase K (Sigma-Aldrich, Saint-Louis, MO, USA) at room temperature before probe hybridization. For each gene, probes were designed by Molecular Instrument (Los Angeles, CA, USA) using the following NCBI accession numbers: XM_038775015.1 (*ScKctd12b*), XM_038811423.1 (*ScProx1*) and XM_038818902.1 (*ScSox1*). The following amplifiers were used respectively for these probes: B3-Alexa Fluor594, B2-Alexa Fluor488 and B1-Alexa Fluor647. DAPI staining, fluorescent IHC of β-catenin (Ab6302, 1/1000, Abcam, Boston, MA, USA), HuC/D (A-21271, 1/400, Molecular Probes) and acetylated tubulin (T-6793, 1/200, Sigma-Aldrich, Saint-Louis, MO, USA) on sections was carried out as previously described[37,39]. For fluorescent ISHs, HCRs, and IHCs, sections were imaged on a Leica SP8 confocal laser-scanning microscope.

### BrdU pulse chase analysis

BrdU (5-bromo-2′-deoxyuridine) pulse labeling and BrdU detection were performed as previously described[39] with the following modifications: catshark embryos were removed from the egg case, incubated in oxygenated filtered sea water containing 5 mg/ml BrdU (16.2 mM) (Sigma-Aldrich, Saint-Louis, MO, USA) for 16 h and, for the chase, transferred to filtered sea water at 16 °C until the desired stages. Following incubation of sections with the anti-BrdU primary antibody (sc-32323, 1/100, Santa Cruz Biotechnology, Dallas, TX, USA), detection was carried out with mouse IgG kappa binding protein conjugated to horseradish peroxidase (sc-516102, 1/100, Santa Cruz Biotechnology, Dallas, TX, USA) and the TSA Plus Cyanine 3 Kit (Akoya Biosciences, Menlo Park, CA, USA) following the supplier's instructions.

### Pharmacological treatments

For Nodal/Activin signaling inhibition, catshark embryos were treated at stage 16 by *in ovo* injection of 100 µl of a 50 µM solution of SB-505124 (Sigma-Aldrich, Saint Louis, MO, USA, a selective inhibitor of TGF-β type I receptors Alk4/5/7/18, as previously described[37]. For Wnt inactivation, 100 µl of a DMSO-based solution containing 1 mM IWR-1 (I0161, Sigma-Aldrich, Saint-Louis, MO, USA), a selective inhibitor of tankyrase known to inhibit the cytoplasmic accumulation of β-catenin through stabilization of the destruction complex member Axin2, were injected into the egg case at stage 29, and eggs were maintained in oxygenated sea water at 16 °C until stage 31. The same protocol was applied to control embryos, except for the absence of the drugs in the injection solutions.

### Reporting summary

Further information on research design is available in the Nature Portfolio Reporting Summary linked to this article.

## Data availability

All data supporting the findings of this study are available within the paper and its Supplementary Information. Transcriptomic data were deposited into the NCBI Sequence Read Archive (SRA) under accession number PRJNA1040760. Novel river lamprey sequences were deposited in the NCBI nucleotide database and are available under numbers PQ241090, PQ241091, PQ241092, PQ241093, PQ241094, PQ241095 and PQ241096.

## Code availability

No custom code or mathematical algorithm was used in this study. Published pipelines used in the transcriptomic analysis are cited in the references.

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

## Acknowledgements

We thank Pascal Romans and the Centre de Ressources Biologiques Marines the Banyuls-sur-Mer Oceanological Observatory (OOB) for help in obtaining specimens, the Sanger Institute for providing access to the catshark genome prior to public release, EMBRC-France for support to local marine infrastructures, David Pecqueur and the BioPic imaging platform for access to confocal microscopy, the Bio2Mar service for access to molecular biology platform, the MGX platform for cDNA library construction and sequencing and the UMR7232 Service de Bio-Informatique BSBII for HM's support. The work was funded by the Centre National de la Recherche Scientifique, by Agence Nationale de la Recherche contract n°ANR-16-CE13-0013-02 to SM (S. Mazan) and PB, and by contract n°ANR-21-CE340006-02 to MS. It was supported by PhD fellowships to ML (LSP n°156393 Région d'Occitanie) and LM (Ministère de la Recherche, ED515).

## Author contributions

SM (S. Marcellini), NP, EC, MDT, CB, MS, PB, and SM (S. Mazan) conceived the study and its experimental design in the different model organisms analyzed; ML, LM, RL, LG, VL, DS, KM, and HC performed experiments; HM, CK and BB performed bioinformatic analyzes; SM (S. Marcellini), MDT, MS and SM (S. Mazan) wrote the manuscript. All authors read and approved the manuscript.

## Competing interests

The authors declare no competing interest.

## Additional information

¹CNRS, Sorbonne Université, UMR7232-Biologie Intégrative des Organismes Marins, Observatoire Océanologique, Banyuls-sur-Mer, France. ²Centre de Ressources Biologiques Marines, Sorbonne Université, Observatoire Océanologique, UMS 2348, Banyuls-sur-Mer, France. ³MGX, Université Montpellier, CNRS, INSERM, Montpellier, France. ⁴UK Research and Innovation, Biotechnology and Biological Sciences Research Council, Swindon, UK. ⁵Plateforme Bioinformatique, Genotoul, BioinfoMics, UR875 Biométrie et Intelligence Artificielle, INRAE, Castanet-Tolosan, France. ⁶Department of Cell Biology, School of Biological Sciences, University of Concepcion, Concepcion, Chile. ⁷Université Paris-Saclay, CNRS, IRD, Évolution, Génomes, Comportement et Écologie, Université Paris-Saclay, Gif-sur-Yvette, France. ⁸Departament of Functional Biology, CIBUS, Faculty of Biology, Universidade de Santiago de Compostela, Santiago de Compostela, Spain. ⁹ISEM, Université de Montpellier, CNRS, IRD, EPHE, Montpellier, France. ¹⁰School of Molecular and Life Sciences, Curtin University, Perth, WA, Australia. ¹¹UMR8227, CNRS-Sorbonne Université, Station Biologique, Roscoff, France. ¹²Laboratoire de Biologie du Développement de Villefranche-sur-Mer, Institut de la Mer de Villefranche, Sorbonne Université, CNRS, Villefranche-sur-Mer, France. ¹³Centre de Biologie Intégrative (CBI, FR 3743), Université de Toulouse, CNRS, UPS, Toulouse, France. ¹⁴These authors contributed equally: Maxence Lanoizelet, Léo Michel, Ronan Lagadec, Hélène Mayeur, Lucile Guichard. ✉e-mail: mazan@obs-banyuls.fr

