## [Peer Review file · Nature Communications]

Analysis of a shark reveals ancient, Wnt-dependent, habenular asymmetries in vertebrates

Corresponding Author: Dr Sylvie Mazan

Version 0:

Reviewer comments:

Reviewer #1

(Remarks to the Author)

The manuscript by Lanoizelet et al, shows that the lateral and medial habenulae in the catshark are arranged in an asymmetric way as in several lower vertebrate species. By using single cell RNAseq they make a detailed analyses of the genes expressed in the medial and lateral habenulae on the left and right side. Moreover, they demonstrate that the asymmetry is due to differences in Wnt signaling on the left and right side respectively. They also show that the same mechanisms account for the asymmetry in the elephant shark, and in the lungfish and the reed fish belonging to separate groups, while the asymmetry has been lost neopterygians (spotted gar and zebrafish) and separately lost in amphibians and mouse belonging to the sarcopterygians (as in lungfish). The conclusion is that this asymmetry occurs in elasmobranchs and some groups of fish but has been lost in others. No attempt was made to investigate if the asymmetry has functional implications, nor the afferent and efferent projection patterns to the different parts of the habenulae. The text is in general well written (see, however, below), and the figures good.

General comments.

- 1) The authors state (line 365) that: "We show here that the catshark habenulae exhibit a highly asymmetric organization, with a majority of the asymmetric gene expression profiles for the first time in a vertebrate". This is simply incorrect. Stephenson-Jones et al (not cited) showed in the lamprey (Evolutionary conservation of the habenular nuclei and their circuitry controlling the dopamine and 5-hydroxytryptophan (5-HT) systems..Proc Natl Acad Sci U S A. 2012) not only that a similar asymmetry occurs in a phylogenetically much older group of vertebrates, a trait that has been conserved in elasmobranchs (cat shark). They further show that the efferent projection pattern of both the medial and lateral habenulae is similar to that of mammals, but also the afferent connectivity to the medial and lateral habenulae. The basic overall organization of the lateral and medial habenulae is thus conserved despite the asymmetry.
- 2) Since, the lamprey has a similar asymmetry with regard to the lateral and medial habenulae and belongs to a phylogenetically much older group of vertebrates it would be appropriate to include the lamprey in the analyses, it would further strengthen the study. Moreover, the general organization of the habenulae is much better known than in any of the species that are included in the study of Lanoizelet. In any case, the authors need to appropriately cite and discuss the data from the lamprey (modify the text in many places), instead of not even citing the data from the lamprey.
- 3) Since Nature journals have a broad readership a very specialized terminology should be avoided or explained. For instance, dipneusts, polypterids, actinopterygians, neopterygeans, sarcopterygians appear in the text without explanation. The latin names for reedfish and lungfish are used intermittently with the common names. This would be simple to remedy in the text.

Specific comments

- 1) Line 1 in the abstract is incorrect (see above). Modify.

Reviewer #2

(Remarks to the Author)

The manuscript by Lanoizelet et al. aims to decipher the evolutionary origin of habenular left-right asymmetry by examining

the molecular anatomy of the catshark habenular nuclei (Hb) in comparison with the zebrafish, mouse and frog brain and with other ancestral fish species. By injecting a chemical antagonist of Wnt signaling in catfish larvae, they further demonstrate that the Wnt pathway promotes neuronal identities characteristic of the right Hb and represses those that are characteristic of the left Hb and, as was shown for the dorsal Hb of zebrafish, seems to regulate the timing of neural proliferation in the developing lateral nuclei. This study is a follow-up and more extensive study to several papers by the same group on left-right asymmetry of the Hb in various fish species. There is an enormous amount of high-quality data provided in this manuscript in 8 main figures and 17 Supplementary figures. However, a main concern is that this work is highly specialized and may not be of interest to a broader audience.

In places, the writing is rather unwieldy and difficult to wade through. Two examples are lines 40-43, "Analysis of the mechanisms underlying their formation highlights an essential role of Wnt signaling, which is submitted to a dynamic, asymmetric regulation during habenula development, with a Nodal-dependent left repression in postmitotic precursors of the lateral habenulae" and lines 94 -97, "We also propose a conserved regulatory logic for asymmetry formation, involving interactions between a conserved temporal control of neurogenesis and a more flexible temporal and spatial regulation of Wnt signaling, which could account for the evolvability of habenular asymmetries across vertebrates." Such dense writing also makes the study less accessible to non-specialists.

Line 60: It is unclear why the authors claim that "more precise comparisons of neuronal populations across vertebrates remain difficult" without providing rationale for this assertion. Indeed, scRNA-seq and bioinformatic analyses provide unprecedented information about neuronal identities.

When the authors use the term "enriched" to describe transcripts in the Hb (e.g., in Figure 1) it is not clear what their reference is, enriched relative to the remaining brain tissue? This is not stated in the methods.

In Line 172-174, it is stated that certain genes known to be asymmetrically expressed in the zebrafish habenulae are "without consistent conservation of asymmetry laterality." The basis for this statement is unclear as the majority of larval zebrafish show conserved directional asymmetry (i.e., >90% bias). There are no references for the above point, so it is unclear how the authors came to this conclusion. Moreover, the identification and expression analyses of the cited genes was performed first in the zebrafish not after the catshark and should be cited as such as well as properly referenced.

The authors conclude that left-right differences between the lateral Hb was an ancestral trait lost in tetrapods; however, there are published reports of asymmetry in this brain region in the human brain. Are these reports not valid? They should at least be mentioned.

Finally, the discussion on the influence of Wnt signaling on neurogenesis in the catshark compared to zebrafish seems somewhat confusing. The results on catshark showed left-right differences but the authors underplay this finding (lines 399-403), yet later argue the opposite (lines 441-444). I found the Discussion not very clear and a bit contradictory.

Minor corrections:

Line 61: A unique feature of the habenulae in the vertebrate brain is that (it) they display (s)

In Supplemental Table 2, the authors credit the Pandey et al. publication for expression analyses of genes (e.g., *kiss1*, *kctd12.1*, *kctd12.2*) in work that was described in earlier papers by others.

Reviewer #3

(Remarks to the Author)

Lanoizelet and colleagues report new insights into the evolution of brain asymmetry and an underlying molecular mechanism analyzing the habenular neural circuit. The zebrafish dorsal habenulae together with the pineal complex serve as model system to study the molecular basis and functional relevance of left-right brain asymmetry. The main signaling pathways involve Nodal, Wnt, Notch and FGF. Furthermore, yet unknown signals from the parapineal influence habenular neurogenesis on the left side in zebrafish. The authors study the evolutionarily distant catshark and find asymmetries in both dorsal as well as ventral/lateral habenular neurogenesis and marker gene expression. Moreover, their investigations show that neurogenesis of the ventral/lateral habenular depends on Wnt/beta-catenin signaling, which in turn is repressed by Nodal signaling on the left hemisphere. Lanoizelet and co-workers further describe habenular marker gene expression in two additional Actinopterygians, three members of the Sarcopterygians and an additional Chondrichthyan. They conclude that conserved molecular building blocks are responsible for the generation of neurons in the habenulo/thalamic area, which show spatio-temporal variations during evolution.

The manuscript by Lanoizelet is well written even if the topic is rather complex. The authors try to improve reader-friendliness by including multiple drawings. However, the manuscript remains difficult to comprehend in particular for non-experts of the field. The conclusions are based on the data obtained. The work is foremost descriptive and the mechanistic data on the molecular level are limited due to the sparse manipulation techniques available to the species used. However, the choice of animals and the experiments carried out are logic and line up nicely. The work will be interesting to a relatively restricted readership of evolutionary, developmental and neuroscientific biologists. I only have a few comments:

- Introduction: The habenulae are also connected to structures in the ventral hindbrain

- Supplementary Fig. 6: Ntng2 (and perhaps also Pcdh17) appears expressed also in the medial habenulae, which is not mentioned in the text (apologies if I missed it).

- Results: It would be nice if the authors could specify more precisely on what basis the developmental stages of animals other than catshark, zebrafish or mouse were chosen for comparison.
- Results, line 242: Supplementary figure 12f is listed to show non-nuclear beta-catenin expression. This seems a mistake.
- The numbers are in general very low. I assume that this is due to the availability of specimens. I could not find numbers for the experiments in Fig. 3g/h – please add.
- Discussion line 408-409: The authors state that Tcf712 is required for kiss1 expression in the ventral habenula of zebrafish. This is not necessarily correct. The authors cite a paper by Beretta and colleagues, who showed that the ventral habenula does not develop in the absence of Tcf712. Therefore, the absence of kiss1 expression might actually be due to the absence of the structure. This should be considered in the argumentation and in the schematic shown.
- Patrick Blader's lab, one of the co-authors of the current manuscript, has previously shown that nodal signaling in zebrafish is essential for early left sided habenular neurogenesis (Roussigne et al., 2009). It would be good to consider and integrate this finding into their hypothesis.

Version 1:

Reviewer comments:

Reviewer #1

(Remarks to the Author)

The manuscript has been further strengthened by the inclusion of the asymmetries observed in cyclostomes. I have no further questions to raise!

Reviewer #2

(Remarks to the Author)

Lanoizelet et al. have included new data on the habenulae of lamprey in addition to their comprehensive analyses of many other species, and have addressed most of the issues raised by the reviewers, with the exception of making their writing more accessible to a broader audience.

Some general comments:

-Line 47: I think the authors should delete this non-informational sentence from the Introduction. It doesn't help to make the presented work of greater interest to a broad audience.
 "From a translational research perspective, it is also important to evaluate the relevance of model organisms for the analysis of neurological disorders in humans."

-Line 56: "Habenular territories have also been implicated in human pathological conditions, such as depression or drug addiction and the understanding of these dysregulations, as inferred from analyses in animal models, implies a robust knowledge of cross-species conservations and divergences". I do not think "implies" is a correct word choice here. Do the authors mean requires or necessitates?
 Either way, I don't follow the logic. There is no real explanation as to why knowledge of cross-species conservation and divergence is useful for understanding depression or drug addiction in humans. This is an unsubstantiated assertion and, again, statements of justification like these do not help make the study of broader interest. We should just accept the fact that the present study is largely descriptive, of high quality and data-rich, but will likely only be of interest to a specialized audience.

This is still a challenging paper to wade through. It might help if the authors refrained from using so many stacked modifiers (e.g., "asymmetric neuronal identity choices of post-mitotic progenitors")

Some suggested corrections:

Line 153: "ScSox1 and ScProx1 expressions are confined" change to "ScSox1 and ScProx1 expression patterns are confined"

Line 173: habenular expression not expressions

Overuse of "highlight" – use show, demonstrate, reveal, depict, etc.

Change "diversifications" to diversification throughout

I agree with the other reviewer that field-specific terms are (still) not adequately described/explained. For example, the authors do not explain the term cyclostome (e.g., jawless vertebrate) on line 395.

Line 553: "This difference is not absolute, as an early neurogenetic asymmetry in the zebrafish is Nodal-dependent (in the zebrafish)" Authors should remove the second "in the zebrafish".

Line 556: "In addition, the molecular programs that repress Wnt activity on the left side respectively downstream Nodal in the catshark but under parapineal influences in the zebrafish remain to be

established” is an awkward sentence that would benefit from punctuation or re-writing for greater clarity. Anyway, line 565 makes the same point but more clearly so there is no need for stating this twice (“Such variations might in turn lay the ground for major mechanistic transitions, such as the switch from a Nodal-dependent left repression of Wnt signaling, as observed in the catshark, to a parapineal-dependent repression, as described in the zebrafish”)

Reviewer #3

(Remarks to the Author)

The authors have addressed all my comments satisfactorily.

ANSWER TO REVIEWER COMMENTS

GENERAL POINTS

We first wish to thank all three reviewers for the time spent on this manuscript and for their constructive comments, both on evolutionary and mechanistic aspects. All have been taken into account in this revised version, which includes additional experimental work and text modifications as requested.

The main experimental additions are the following:

- (i) We have added the lamprey in the comparative analysis, focusing on the same candidate genes as those analyzed in the elephant shark, the reedfish, the spotted gar, the lungfish and the frog. This results in a more complete picture of the evolution of habenular asymmetries, which reveals putative ancestral vertebrate habenular asymmetries, but also major differences between cyclostomes (agnathans) and gnathostomes (jawed vertebrates).
- (ii) We have expanded the number of catshark embryos analyzed in pharmacological treatments, which confirms the phenotypes we previously observed.
- (iii) We now report asymmetric *in situ* hybridization signals for *Kiss1*, in addition to *Prox1*, in lungfish right lateral habenulae (Figure 2v; Supplementary Figure 10h). This point was not requested by the reviewers, but it significantly strengthens the conservation of lateral habenula asymmetries in a species occupying a key phylogenetic position in sarcopterygians. The ancestral nature of this trait, as proposed in the manuscript, is further supported by this result.

The main changes in the text concern:

- (i) We have placed the work in the broader context of cerebral diversifications during vertebrate evolution and explained why habenular asymmetries are a unique system to address this general question.
- (ii) We have clarified our discussion on the evolution of the mechanisms controlling asymmetry formation and on the involvement of asymmetric neurogenesis, with a clearer distinction between lateral and medial habenulae.

A more detailed description of all changes in the revised manuscript is provided in the point-by-point answer below (blue characters). We hope that they address the reviewers' concerns and improve the manuscript to the point that it can meet the high standard required for publication in Nature Communications.

POINT-BY-POINT ANSWER

1. Reviewer #1 (Remarks to the Author):

The manuscript by Lanoizelet et al, shows that the lateral and medial habenulae in the catshark are arranged in an asymmetric way as in several lower vertebrate species. By using single cell RNAseq they make a detailed analyses of the genes expressed in the medial and lateral habenulae on the left and right side. Moreover, they demonstrate that the asymmetry is due to differences in Wnt signaling on the left and right side respectively. They also show that the same mechanisms account for the asymmetry in the elephant shark, and in the lungfish and the reed fish belonging to separate groups, while the asymmetry has been lost neopterygians (spotted gar and zebrafish) and separately lost in amphibians and mouse belonging to the sarcopterygians (as in lungfish). The conclusion is that this asymmetry occurs in elasmobranchs and some groups of fish but has been lost in others. No attempt was made to investigate if the asymmetry has functional implications, nor the afferent and efferent projection patterns to the different parts of the habenulae. The text is in general well written (see, however, below), and the figures good.

We thank the reviewer for this assessment and the suggestion to expand the work to the lamprey (see below). We also fully agree that direct analyses of projection patterns and of asymmetry functions are important points to understand the biological significance of asymmetries and of their diversification. We chose to focus here on molecular characterizations and mechanistic aspects, but we now explicitly indicate these perspectives at the end of the main text.

Discussion, lines 567-570: *"How such developmental changes, affecting a morphological trait known to regulate important organismal responses to environmental cues, may correlate with diversifications of neuronal circuitry and of behavioral adaptations in different ecological contexts, will be an important challenge for the future".*

General comments.

1.1. The authors state (line 365) that: "We show here that the catshark habenulae exhibit a highly asymmetric organization, with a majority of the asymmetric gene expression profiles for the first time in a vertebrate". This is simply incorrect.

We meant to say that most of the genes identified as asymmetrically expressed in the catshark are reported as such for the first time in a vertebrate, but the original wording was indeed ambiguous. We have clarified this point as follows.

Discussion, lines 435-437: *"We show that catshark habenulae exhibit a highly asymmetric organization, with many of the genes identified here as asymmetrically expressed in this species being reported as such for the first time in a vertebrate".*

1.2. Stephenson-Jones et al (not cited) showed in the lamprey (Evolutionary conservation of the habenular nuclei and their circuitry controlling the dopamine and 5-hydroxytryptophan (5-HT) systems. Proc Natl Acad Sci U S A. 2012) not only that a similar asymmetry occurs in a phylogenetically much older group of vertebrates, a trait that has been conserved in elasmobranchs (cat shark). They further show that the efferent projection pattern of both the medial and lateral habenulae is similar to that of mammals, but also the afferent connectivity to the medial and lateral habenulae. The basic overall organization of the lateral and medial habenulae is thus conserved despite the asymmetry.

While the previous version of the manuscript focused on gnathostomes, we now include the lamprey in our analysis, following the reviewer's recommendation. We describe the asymmetries reported in this species in the introduction and cite corresponding references.

Introduction, lines 83-91: *"In the river lamprey, a member of the cyclostomes (agnathans), which diverged from gnathostomes, their sister group in vertebrates, more than 500 million years ago (Mya)²⁸, analyses of afferent and efferent projections highlighted the conservation of the bipartite organization of habenulae. However, they suggested different relative locations of territories homologous to the dorsal/medial and ventral/lateral components of gnathostomes, and a highly distinctive asymmetry pattern. Putative lamprey homologs of the ventral/lateral habenula of gnathostomes indeed map exclusively to the right habenula, while those of the dorsal/medial habenula are distributed between both sides, comprising the entire left habenula, and a smaller, right-restricted, medial cell nucleus^{29,30}".*

We also indicate the mechanistic divergence previously reported in this species for asymmetry formation relative to the zebrafish.

Introduction, lines 98-102: *"Analyses of a shark and a lamprey, both endowed with marked habenular asymmetries, highlight a major difference with the zebrafish. While the involvement of Wnt signaling remains an opened question in these species, they both require an early left-restricted diencephalic activity of Nodal, dispensable in the zebrafish, for habenular asymmetry formation³⁷".*

Finally, we more clearly indicate why the recurrent presence of asymmetries, in a number of vertebrate phyla including early diverging ones, does not necessarily imply a single, ancient origin in the taxon.

Introduction, lines 102-105: *"In view of these variations, it is still unclear whether habenular asymmetries arose several times independently in the different vertebrate lineages, possibly due to common developmental constraints, or whether they diversified from an ancestral pattern, differentially modified in individual lineages".*

1.3. Since, the lamprey has a similar asymmetry with regard to the lateral and medial habenulae and belongs to a phylogenetically much older group of vertebrates it would be appropriate to include the lamprey in the analyses, it would further strengthen the study. Moreover, the general organization of the habenulae is much better known than in any of the species that are included in the study of *Lanoizelet*. In any case, the authors need to appropriately cite and discuss the data from the lamprey (modify the text in many places), instead of not even citing the data from the lamprey.

We thank the reviewer for this suggestion to address the origin of habenular asymmetries in vertebrates rather than only in gnathostomes, which significantly expands the significance of the work. We have now included a cyclostome, the river lamprey, in our comparative analysis, and the manuscript has been expanded as follows.

(i) We describe the subdomain organization and asymmetry pattern observed in the lamprey in an additional main figure (Figure 8). The corresponding phylogenetic analyses and the selection of paralogs submitted to *in situ* hybridization experiments have been added as Supplementary Figure 18 and Supplementary Table 5 and a novel paragraph (**Results, lines 391-433**) of the Results section entitled *"Gnathostome asymmetries in lateral habenulae are partially conserved in the river lamprey, with a different habenular architecture"*. Briefly, we find that the results are consistent with previously reported afferent and efferent projection patterns and that they highlight both conservation and divergence within gnathostomes.

(ii) We have included these conclusions in the discussion.

Discussion, lines 456-466: *"Comparisons at a wider evolutionary scale, between the catshark and the river lamprey, also reveal similarities in asymmetry patterns, with the restriction to the right side of a broad territory related to the catshark lateral right habenula and a bilateral distribution of territories expressing members of the Kctd8/12/12b/16 family. However, we found no evidence for habenular territories related to the catshark left lateral habenula, with the whole lamprey left habenula expressing LfKctd12l. Similarly, contrary to gnathostomes, we observed several nuclei of mixed identity, co-expressing LfKctd12l and a Prox1 family member. The organization of the lamprey habenulae emerging from these molecular comparisons, although based only on a limited number of markers, is consistent with a projection analysis, which highlighted a right restriction of territories related to the gnathostome lateral habenula and a circuitry reminiscent of those involving the gnathostome medial habenula on the left^{29,30}".*

(iii) We have expanded the evolutionary scenario, which we propose for the diversification of habenular asymmetries accordingly.

Figure 9a, node 1 and lines 466-471: *"Taken together, these data suggest a multi-step mode of habenular asymmetry evolution involving (1) an ancient restriction in vertebrates of neuronal lateral right identities to the right side in habenulae already endowed, as previously proposed²⁹, with a bipartite organization, (2) the fixation, early in the gnathostome lineage, of bilateral medial and lateral subdomains, in the relative topological organization observed in all extant gnathostomes, and of lateral habenula asymmetries related to those reported in the catshark,..."*

(iv) We now discuss remaining questions as to the vertebrate ancestral state in view of the differences observed between the lamprey and gnathostomes.

Discussion, lines 485-488: *"This evolutionary scenario leaves several questions unanswered. First, the organization and the asymmetries of lamprey habenulae may equally reflect the vertebrate ancestral state, or a derived condition from an ancient gnathostome-like pattern. This point could be clarified by analyses of hagfishes, the sister group of lampreys in cyclostomes."*

1.4. Since Nature journals have a broad readership a very specialized terminology should be avoided or explained. For instance, dipneusts, polypterids, actinopterygians, neopterygeans, sarcopterygians appear in the text without explanation.

We have systematically defined phyla that may be unfamiliar to some readers, and we have added references explaining why their study is important for understanding vertebrate evolution.

Introduction, lines 74-75: *"..., a partitioning conserved in gnathostomes (jawed vertebrates)^{16-19"}*;

Introduction, lines 83-85: *"In the river lamprey, a member of the cyclostomes (agnathans), which diverged from gnathostomes, their sister group in vertebrates, more than 500 million years ago (Mya)²⁸,..."*;

Results, lines 208-217: *"Our species sampling included representatives of the three main gnathostome phyla: chondrichthyans, actinopterygians (ray-finned fishes) and sarcopterygians (lobe-finned fishes and tetrapods). We selected five species for their phylogenetic position within these phyla: the elephant shark *Callorhynchus milii* (chondrichthyan, member of holocephalans, the sister group of elasmobranchs⁴⁶), two non-teleostean actinopterygians, the reedfish *Erpetoichthys calabaricus* (member of polypterids, the earliest diverging lineage in actinopterygians⁴⁷) and the spotted gar *Lepisosteus oculatus* (member of holosteans, which together with teleosts, their sister group, form neopterygians⁴⁸), and two sarcopterygians, the African lungfish *Protopterus annectens* (member of lungfishes, the closest living relatives to tetrapods⁴⁹) and the Western clawed frog, *Xenopus tropicalis*."*

The latin names for reedfish and lungfish are used intermittently with the common names. This would be simple to remedy in the text.

We have fixed this incongruency throughout the manuscript: Latin names are only cited, together with common names, the first time when we mention the corresponding species. We then systematically use common names. As an exception, we have nevertheless maintained both names in the Methods section.

1.5. Specific comments

Line 1 in the abstract is incorrect (see above). Modify.

We think that the previously documented presence of extensively divergent habenular asymmetries across vertebrates, including early diverging lineages such as cyclostomes, left their evolutionary origin unclear, as mentioned in the previous version. But we have now clarified what we referred to in the introduction.

Introduction, lines 102-105 : *"In view of these variations, it is still unclear whether habenular asymmetries arose several times independently in the different vertebrate lineages, possibly due to common developmental constraints, or whether they diversified from an ancestral pattern, differentially modified in individual lineages."*

We also understand that our previous wording in the abstract was too restrictive, focusing solely on the evolutionary origin of habenular asymmetries, while we also provide insights into evolutionary trends in different vertebrate lineages. We have therefore replaced the first sentence of the abstract by the following:

Abstract, lines 29-30: *"The mode of evolution of left-right asymmetries in the vertebrate habenula remains largely unknown."*

2. Reviewer #2 (Remarks to the Author):

The manuscript by Lanoizelet et al. aims to decipher the evolutionary origin of habenular left-right asymmetry by examining the molecular anatomy of the catshark habenular nuclei (Hb) in comparison with the zebrafish, mouse and frog brain and with other ancestral fish species. By injecting a chemical antagonist of Wnt signaling in catfish larvae, they further demonstrate that the Wnt pathway promotes neuronal identities characteristic of the right Hb and represses those that are characteristic of the left Hb and, as was shown for the dorsal Hb of zebrafish, seems to regulate the timing of neural proliferation in the developing lateral nuclei. This study is a follow-up and more extensive study to several papers by the same group on left-right asymmetry of the Hb in various fish species. There is an enormous amount of high-quality data provided in this manuscript in 8 main figures and 17 Supplementary figures.

We thank the reviewer for this assessment of our work.

2.1. However, a main concern is that this work is highly specialized and may not be of interest to a broader audience.

To take this point into account, we have placed the work in a broader framework, focusing on how the vertebrate brain diversifies during evolution. We explain why this question is important to understand behavioral adaptations, but also to assess the relevance of animal models to the study of neurological pathologies. We also mention why habenular asymmetries are a unique system for this question from an evolutionary developmental perspective.

Introduction, lines 46-68: *"Knowledge on how cerebral structures diversify during evolution is crucial for understanding how animals adapt their behavioral strategies to varied environmental challenges. From a translational research perspective, it is also important to evaluate the relevance of model organisms for the analysis of neurological disorders in humans. The vertebrate habenula presents unique characteristics to address this general question. This bilateral epithalamic structure forms a relay in conserved neuronal circuits connecting various forebrain areas to midbrain and brainstem nuclei. It appears as a key node, which integrates information from multiple sources (sensory organs, corticolimbic areas), and regulates a variety of adaptive behavioral, cognitive and emotional responses¹⁻⁶. The neuronal basis for these regulations is a field of intense investigations in the mouse and the zebrafish, but their conservation remains largely unexplored in other species and different ecological contexts. Habenular territories have also been implicated in human pathological conditions, such as depression or drug addiction^{7,8}, and the understanding of these dysregulations, as inferred from analyses in animal models, implies a robust knowledge of cross-species conservations and divergences. Finally, a remarkable feature of habenulae is that in many vertebrates, they display asymmetries between their left and their right sides. Their biological significance has started to emerge in the zebrafish, the reference model for their analysis⁹. In this species, habenular asymmetries result in a differential integration of sensory cues between the right and the left habenulae, and they impact important adaptive responses, such as exploratory and food-seeking behaviors, light preference or responses to fear¹⁰⁻¹³. Besides the evolutionary significance of these functions, this trait is highly significant from an evolutionary developmental biology perspective, as the generation of neuronal diversity both during ontogeny, between the two sides of a same, functional, structure, and during evolution, depending on ecological contexts, paves the way for explorations of connections between these two levels."*

The perspectives opened by the work are also mentioned in the last sentence of the discussion.

Discussion, lines 567-570 : *"How such developmental changes, affecting a morphological trait known to regulate important organismal responses to environmental cues, may correlate with diversifications of neuronal circuitry and of behavioral adaptations in different ecological contexts, will be an important challenge for the future."*

2.2. In places, the writing is rather unwieldy and difficult to wade through. Two examples are lines 40-43, “Analysis of the mechanisms underlying their formation highlights an essential role of Wnt signaling, which is submitted to a dynamic, asymmetric regulation during habenula development, with a Nodal-dependent left repression in postmitotic precursors of the lateral habenulae” and lines 94-97, “We also propose a conserved regulatory logic for asymmetry formation, involving interactions between a conserved temporal control of neurogenesis and a more flexible temporal and spatial regulation of Wnt signaling, which could account for the evolvability of habenular asymmetries across vertebrates.” Such dense writing also makes the study less accessible to non-specialists.

We have largely re-written the corresponding part of the abstract, avoiding long sentences and complex expressions. When doing so, we have also taken into account point 2.6 below, clarifying the fact that we observe distinct mechanisms of asymmetry formation in lateral and medial habenulae, only those in the medial habenulae involving an asymmetric temporal control of neurogenesis.

Abstract, lines 34-44: *"Asymmetry formation involves distinct mechanisms in the catshark lateral and medial habenulae. Medial habenulae are submitted to a marked, asymmetric temporal regulation of neurogenesis, undetectable in their lateral counterparts. Conversely, asymmetry formation in lateral habenulae results from asymmetric neuronal identity choices of post-mitotic progenitors, a regulation dependent on the repression of Wnt signaling by Nodal on the left. Based on comparisons with the mouse and the zebrafish, we propose that habenular asymmetry formation involves a recurrent developmental logic across vertebrates, which relies on conserved, temporally regulated genetic programs sequentially shaping neuronal identity choices on both sides and asymmetrically modified by Wnt activity. The functional pleiotropy of the Wnt pathway and the dynamic nature of its regulation, prone to variations in time and space, may be major driving forces for asymmetry diversification during vertebrate evolution."*

2.3. Line 60: It is unclear why the authors claim that “more precise comparisons of neuronal populations across vertebrates remain difficult” without providing rationale for this assertion. Indeed, scRNA-seq and bioinformatic analyses provide unprecedented information about neuronal identities.

What we meant by this sentence is that apart from correspondence between the mouse medial and lateral habenulae and their zebrafish dorsal and ventral counterparts, it is difficult to identify relationships between the discrete neuronal populations revealed by single-cell RNA-seq analyses in these two species, despite the extremely detailed information, which they contain. The differential evolutionary trends highlighted by comparisons with the catshark (the mouse habenulae being more similar to the catshark left habenulae, unlike the zebrafish ventral habenulae, more similar to the catshark right lateral habenulae) likely contribute to this difficulty, because they might obscure ancestral traits. However, we understand that this hypothesis remains very speculative in the absence of more detailed data in the catshark, including single-cell RNA-seq data, and we have therefore removed this sentence from the revised manuscript.

2.4. When the authors use the term “enriched” to describe transcripts in the Hb (e.g., in Figure 1) it is not clear what their reference is, enriched relative to the remaining brain tissue? This is not stated in the methods.

We have clarified this point in the Methods section. Left- or right-enriched refers to a statistically significant expression difference relative to the contralateral side.

Methods, lines 600-603: *"Contigs exhibiting statistically significant count differences between the left and right habenulae were identified using the Wald test (q-value threshold 5E-02) implemented in sleuth⁶⁵. We referred to these genes as left- or right-enriched, by comparison to the contralateral side."*

2.5. In Line 172-174, it is stated that certain genes known to be asymmetrically expressed in the zebrafish habenulae are “without consistent conservation of asymmetry laterality.” The basis for this statement is unclear as the majority of larval zebrafish show conserved directional asymmetry (i.e., >90% bias). There are no references for the above point, so it is unclear how the authors came to this conclusion.

Throughout the manuscript, we take a strict evolutionary terminology and we systematically refer to cross-species comparisons when using terms such as "conserved" or "conservation". We therefore do not refer to intraspecific laterality variations in the zebrafish here. To remove this ambiguity in this sentence, we have specified the fact that we refer to comparisons between the zebrafish and the catshark.

Results, lines 191-194: *"Orthologs of three asymmetrically expressed catshark MHB markers (Spon1, Kctd8, and Kctd12a) display asymmetric expressions in the zebrafish dHb, albeit without consistent conservation of asymmetry laterality between the two species."*

Moreover, the identification and expression analyses of the cited genes was performed first in the zebrafish not after the catshark and should be cited as such as well as properly referenced.

We have put more emphasis in the text on the fact that in this first step of the comparative analysis, our strategy relies on a comparison of the asymmetries validated in the catshark with previously published profiles in the mouse and in the zebrafish, whether asymmetric or not.

Results, lines 171-175: *"To characterize the level of conservation of habenular asymmetry and subdomain organization across gnathostomes, we initially focused on comparisons of the catshark with the mouse and the zebrafish. For this analysis, we systematically surveyed previously published habenular expressions of mouse and zebrafish orthologs of markers of the five major habenular territories identified in the catshark, including Kctd8/12a/12b paralogous genes"*

We have further included, in the main text, references previously only cited in Supplementary Table 2 and we have added missing references for the zebrafish, as well as for the mouse, in the main text and in the legend of Supplementary Table 2. We hope that these modifications give due credit to previous studies of colleagues working on reference model organisms.

Results, lines 176-177: *"In the mouse, this analysis was based on profiles published in the habenulae⁴⁰⁻⁴³, including systematic searches in the Allen Brain Atlas⁴⁴."*

Results, lines 188-191: *"For comparisons with the zebrafish, we examined the presence of orthologs of catshark territory markers in gene signatures of single-cell RNA-seq gene clusters mapped to larval or adult habenulae¹⁹, in addition to previously published ISH data^{20-22,45} (Supplementary Table 2)."*

2.6. The authors conclude that left-right differences between the lateral Hb was an ancestral trait lost in tetrapods; however, there are published reports of asymmetry in this brain region in the human brain. Are these reports not valid? They should at least be mentioned.

We now address this point in the discussion, as part of unresolved questions. We first recall the absence of obvious relationships between the marked asymmetries observed in catshark lateral habenulae and suggest different evolutionary and developmental origins of the subtle asymmetries detected in mammals.

Discussion, lines 494-497: *"Finally, there is also no evidence for a relationship between the marked molecular asymmetries reported here in the catshark and the subtle ones recently reported in mammalian habenulae, including their lateral component^{15,26,27}, which may have completely different evolutionary and developmental origins."*

2.7. Finally, the discussion on the influence of Wnt signaling on neurogenesis in the catshark compared to zebrafish seems somewhat confusing. The results on catshark showed left-right

differences but the authors underplay this finding (lines 399-403), yet later argue the opposite (lines 441-444). I found the Discussion not very clear and a bit contradictory.

We thank the reviewer for this comment, which led us to clarify this important point in the text and to enrich the discussion. This apparent contradiction is related to the difference in the mechanism of asymmetry formation between catshark lateral and medial habenulae, which were respectively described in lines 399-403 and lines 441-444 of the previous manuscript. We find no evidence for an asymmetric temporal regulation of neurogenesis in the developing catshark lateral habenula. In this component, our data also support the conclusion that Wnt signaling shapes the elaboration of right neuronal identities in post-mitotic neurons. In contrast, in the catshark medial habenula, we report earlier cell cycle exits on the left than on the right in the external subterritory, which is reminiscent of the situation described in the zebrafish. In our treatment experiments (conducted relatively late, at a stage when lateral habenula progenitors are post-mitotic), we could not highlight a role for Wnt signaling in shaping this neurogenetic asymmetry (contrary to the zebrafish), but the differential accumulation of β -catenin observed between asymmetric territories of the external medial habenula suggests an asymmetric regulation of the Wnt pathway in this habenular component.

We have now clarified these points in the discussion as follows.

(i) We now completely separate the discussion of mechanisms in the lateral habenula (**Discussion, lines 499-522**) and in the medial habenula (**Discussion, lines 523-545**).

(ii) We emphasize the mechanistic difference between the catshark lateral habenula and the medial one.

Discussion, lines 523-527: *"Concerning asymmetries in the medial/dorsal habenulae, even though their mode of diversification remains unclear, their formation may also involve conserved mechanisms. In this case, however, the underlying cellular mechanism might be an asymmetric temporal control of neurogenesis, as described in the zebrafish^{32,36,45}, rather than an asymmetric control of cell fate choices in post-mitotic neurons as observed in catshark lateral habenulae."*

(iii) We also make this point very clear in the abstract.

Abstract, lines 34-38: *"Their formation involves distinct mechanisms in lateral and medial territories. The medial habenulae are submitted to a marked, asymmetric temporal regulation of neurogenesis, undetectable in their lateral counterparts. Conversely, asymmetry formation in the lateral habenulae results from asymmetric neuronal identity choices of post-mitotic progenitors, a regulation dependent on Wnt signaling, repressed by Nodal on the left."*

(iv) Finally, based on the different cellular functions observed or reported for Wnt signaling in habenular asymmetry formation, we introduce the hypothesis that Wnt functional pleiotropy may be a factor contributing to asymmetry diversification during evolution.

Abstract, lines 38-44: *"Based on comparisons with the mouse and the zebrafish, we propose that habenular asymmetry formation involves a recurrent developmental logic across vertebrates, which relies on conserved, temporally regulated genetic programs sequentially shaping neuronal identity choices on both sides and asymmetrically modified by Wnt activity. The functional pleiotropy of the Wnt pathway and the dynamic nature of its regulation, prone to variations in time and space, may be major driving forces for asymmetry diversification during vertebrate evolution."*

Discussion, lines 561-565: *"On a morphological level, in a structure forming through sequential neurogenesis and neuronal differentiation processes such as the habenulae, the highly dynamic regulation of Wnt signaling, prone to variations in time and space, and the pleiotropy of its effects on different successive progenitor states, are substrates for diversifications of asymmetries, as suggested in teleosts⁵⁶."*

2.8. Minor corrections:

Line 61: A unique feature of the habenulae in the vertebrate brain is that (it) they display (s)

This correction has been implemented.

Introduction, lines 59-60: *"Finally, a remarkable feature of habenulae is that in many vertebrates, they display asymmetries between their left and their right sides."*

In Supplemental Table 2, the authors credit the Pandey et al. publication for expression analyses of genes (e.g., *kiss1*, *kctd12.1*, *kctd12.2*) in work that was described in earlier papers by others.

Missing references have been added in the legend of Supplementary Table 2 and all references cited in this table have also been included in the main text (see point 2.5 above).

Reviewer #3 (Remarks to the Author):

Lanoizelet and colleagues report new insights into the evolution of brain asymmetry and an underlying molecular mechanism analyzing the habenular neural circuit. The zebrafish dorsal habenulae together with the pineal complex serve as model system to study the molecular basis and functional relevance of left-right brain asymmetry. The main signaling pathways involve Nodal, Wnt, Notch and FGF. Furthermore, yet unknown signals from the parapineal influence habenular neurogenesis on the left side in zebrafish. The authors study the evolutionarily distant catshark and find asymmetries in both dorsal as well as ventral/lateral habenular neurogenesis and marker gene expression. Moreover, their investigations show that neurogenesis of the ventral/lateral habenular depends on Wnt/beta-catenin signaling, which in turn is repressed by Nodal signaling on the left hemisphere. Lanoizelet and co-workers further describe habenular marker gene expression in two additional Actinopterygians, three members of the Sarcopterygians and an additional Chondrichthyan. They conclude that conserved molecular building blocks are responsible for the generation of neurons in the habenulo/thalamic area, which show spatio-temporal variations during evolution.

3.1. The manuscript by Lanoizelet is well written even if the topic is rather complex. The authors try to improve reader-friendliness by including multiple drawings. However, the manuscript remains difficult to comprehend in particular for non-experts of the field. The conclusions are based on the data obtained. The work is foremost descriptive and the mechanistic data on the molecular level are limited due to the sparse manipulation techniques available to the species used. However, the choice of animals and the experiments carried out are logic and line up nicely.

We thank the reviewer for this detailed report, taking into consideration the species sampling and the experimental limitations inherent to the use of non-conventional model organisms, whose study is nevertheless essential to test evolutionary scenarios at a macroevolutionary scale.

The work will be interesting to a relatively restricted readership of evolutionary, developmental and neuroscientific biologists.

This point was also noted by reviewer 2 and to take it into account, we have placed the work in a broader context, explaining the importance of understanding how cerebral structures diversify during evolution and the unique features of habenulae for this analysis. The corresponding changes in the text are detailed in point 2.1 above.

To address the complexity of the manuscript for non-experts of the field, we have simplified several figures.

Figure 1: we have removed horizontal sections, which were redundant with Supplementary Fig. 4.

Figures 3 and 4: we have reorganized the panels to align lateral left and right habenula markers on the right and on the left, respectively.

Main text: As suggested by reviewer 1, we have also simplified species names using common terminologies rather than Latin names. We now define the taxa studied and briefly explain why their study is important in a phylogenetic context (changes detailed in point 1.3 above). Finally, the discussion of the mechanisms involved in asymmetry formation in catshark lateral and medial habenulae has been clarified, following a comment of reviewer 2 (changes detailed in point 2.7 above).

I only have a few comments:

3.2. Introduction: The habenulae are also connected to structures in the ventral hindbrain

We have modified the text accordingly.

Introduction, lines 50-51: *"This bilateral epithalamic structure forms a relay in conserved neuronal circuits connecting various forebrain areas to midbrain and brainstem nuclei."*

3.3. Supplementary Fig. 6: Ntng2 (and perhaps also Pcdh17) appears expressed also in the medial habenulae, which is not mentioned in the text (apologies if I missed it).

We now detail the signals observed for mouse orthologs of catshark left lateral habenula markers, including medial habenula and thalamic territories. The only territory shared by all three genes is the lateral habenula.

Results, lines 180-183: *"Two of the four gene signatures of the catshark Left-LHb (Ntng2 and Pcdh17) share bilateral expression in mouse lateral habenulae, with additional signals in the medial habenula (Ntng2) and in the paraventricular nucleus of the thalamus (PVT), adjacent and ventral to the lateral habenulae (Pcdh17) (Supplementary Fig.6e-f)."*

3.4. Results: It would be nice if the authors could specify more precisely on what basis the developmental stages of animals other than catshark, zebrafish or mouse were chosen for comparison.

This choice was highly constrained by the availability of specimens in these species, most of which are only accessible at late embryonic or juvenile stage, and we now explicitly indicate this points (

Results, lines 217-220: *"In each one of these species, we carried out ISH analyses on transverse sections of habenulae at stages when specimens were available (elephant shark: pre-hatching embryos; reedfish, spotted gar, lungfish: juveniles; frog: stage NF-66 tadpoles), using probes..."*

3.5. Results, line 242: Supplementary figure 12f is listed to show non-nuclear beta-catenin expression. This seems a mistake.

This Supplementary Figure is focused on the medial habenula, where we never detected nuclear β -catenin signals (contrary to the right lateral habenula, as detailed in Figure 3). However, we observe a heterogeneity of cytoplasmic β -catenin signals in the medial habenulae, selectively absent in the left-restricted anterior territory of the left external MHB at stage 31. We have emphasized this point in the text. We now also clearly delineate medial habenula territories in Supplementary Figure 12 (dashed lines for the MHB and dotted lines for the left-restricted anterior subdomain of the external MHB in Supplementary Figure 12). These changes have also been introduced in Supplementary Figure 17, which describes β -catenin signals in habenulae of catshark juveniles, with the same conclusions.

Results, lines 263-271: *"This asymmetry is maintained at stage 31, with a high proportion of positive nuclei in the Prox1-expressing Right-LHb and a complete absence of signal (neither nuclear, nor cytoplasmic) in the Sox1-positive Left-LHb (Fig.3e, 3f; compare Fig.3e1-e2 with Fig.3e3-e4). In the Kctd12b-positive MHB (Supplementary Fig.12a,c,e), contrary to to the right lateral habenula, the β -catenin signal is excluded from nuclei in all areas examined (see MHB signals in Supplementary Fig.12b,d,f,b1-b2,d1-d8). However, a heterogeneity is observed, with*

cytoplasmic β -catenin expressions detected in all MHB subdomains except in the anterior left-restricted ScPde1a-positive external component (Supplementary Fig.12b,d; compare Supplementary Fig.12d1-d2 to d3-4, d5-6 and d7-d8)."

Results, lines 368-372: *"In the catshark, IHC profiles of β -catenin remain highly asymmetric in the habenulae of juveniles (Supplementary Fig.17), with nuclear signals restricted to the lateral right habenula (Supplementary Fig.17a7,a8,b3,b4,c5,c6). As at stage 31, heterogeneities are also observed in the medial habenula, with cytoplasmic signals being selectively absent in its external anterior left component (Supplementary Fig.17a,a3,a4)."*

3.6 The numbers are in general very low. I assume that this is due to the availability of specimens.

We thank the reviewer for considering the difficulty of experimental analyses in a shark. The availability of embryos is indeed a limiting factor for experimental analyses in the catshark, due to (i) the seasonality of reproduction, (ii) the low reproduction rate (about 2 eggs per female every two weeks during the spring, which corresponds to the peak of the spawning season) and (iii) the relatively large size of adults, precluding the maintenance of large breeder cohorts in laboratory facilities.

However, we have substantially increased the number of embryos analyzed, in both IWR-1 and double SB505124+IWR-1 treatments with, respectively, 9 and 10 embryos analyzed. We have also increased the number of injection control embryos (DMSO-injected), albeit to a lesser extent (8 and 6 embryos analyzed, respectively), since we never observed abnormalities in these embryos relative to uninjected control embryos. Numbers have been detailed for each marker tested in **Supplementary Tables 3 and 4**.

Results, lines 291-294: *"Expression of the two Left-LHb markers ScSox1 and ScNtng2 expands to the right in all embryos injected with the drug (n=9/9; Supplementary Table 3), a phenotype never observed in control embryos (n=8/8; compare Fig.4b,d,f and Fig.4i,k,m; Supplementary Fig.13b,d1,f,h1)."*

Results, lines 320-328: *"As expected, in ovo injections of SB-505124 consistently results in a right isomerism (n=6/6), with a symmetric nuclear distribution of β -catenin in both left and right lateral habenulae (Fig.5a; Supplementary Fig.15a), a complete loss of lateral left expression of the two Left-LHb markers ScSox1 and ScNtng2 (Fig.5b,d,f; Supplementary Fig.15b,c) and a concomitant lateral left expansion of the Right-LHb markers ScProx1 and ScKiss1 (Fig.5c,e,g; Supplementary Fig.15d). Following IWR-1 injection at stage 29 of SB-505124-treated embryos (n=10/10), habenular symmetry is consistently maintained, but lateral habenulae exhibit non-overlapping territories of Right- and Left-LHb identities, similar to the right habenulae of IWR-1-treated embryos (Fig.5h-n; Supplementary Fig.15e-h)."*

I could not find numbers for the experiments in Fig. 3g/h – please add.

The number of embryos treated with a single injection of DMSO or of SB-505124 (prior to the early diencephalic window of Nodal) was restricted to 2 each, as this experiment merely confirmed the right isomerism that was previously reported in a different study (Lagadec et al. 2015. Nature Com. 6:6686). We did not add a table for this analysis, since a single marker (β -catenin) was analyzed, but the numbers are indicated directly on the panels in **Figure 3g,h** ("n=2/2").

3.7. Discussion line 408-409: The authors state that Tcf712 is required for kiss1 expression in the ventral habenula of zebrafish. This is not necessarily correct. The authors cite a paper by Beretta and colleagues, who showed that the ventral habenula does not develop in the absence of Tcf712. Therefore, the absence of kiss1 expression might actually be due to the absence of the structure. This should be considered in the argumentation and in the schematic shown.

This is an excellent point and we have modified the text accordingly. **Figure 9b** and the corresponding legend have also been modified to take this point into account.

Discussion, lines 506-509: *"Whether a similar Wnt-dependent regulation of neuronal identities in post-mitotic neurons takes place in the ventral habenula of zebrafish, molecularly related to the catshark lateral right habenula, remains to be assessed, since the loss of this territory in tcf712 mutants precludes an easy dissection of the successive functions of this gene in this species³³."*

Legend to Figure 9b, lines 830-832: *"In the zebrafish, Wnt signaling is required for ventral habenula formation, and possible later roles of the Wnt pathway in the elaboration of neuronal identities in this territory remain to be assessed."*

3.8. Patrick Blader's lab, one of the co-authors of the current manuscript, has previously shown that nodal signaling in zebrafish is essential for early left sided habenular neurogenesis (Roussigne et al., 2009). It would be good to consider and integrate this finding into their hypothesis.

We have followed this suggestion and integrated this point in the discussion, with a reference to Roussigné et al. 2009. We propose that the early Nodal-dependent neurogenetic asymmetry detected in the zebrafish may reflect a vestigial trait from an ancestral regulation by Nodal, a hypothesis already put forward in one of our previous publications (Lagadec et al. 2015. Nat. Commun. 6:6686), and now further supported by the detection of an asymmetric control of neurogenesis in the catshark as in the zebrafish.

Discussion, lines 546-556: *"In summary, despite an extensive divergence of habenular asymmetries between the zebrafish and the catshark, their mechanisms of formation display remarkable similarities, which likely reflect ancestral traits. Those include an asymmetric temporal regulation of neurogenesis in medial/dorsal habenulae, a highly dynamic regulation of Wnt signaling and a left-restricted repression of the Wnt pathway, resulting in asymmetric modifications of neuronal identities, albeit via different cellular mechanisms in medial and lateral habenular contexts. However, a major difference is that Wnt signaling is submitted to different asymmetric regulations in the two species, respectively Nodal-dependent in the catshark and parapineal-dependent in the zebrafish^{35,37}. This difference is not absolute, as an early neurogenetic asymmetry in the zebrafish is Nodal-dependent in the zebrafish⁶¹, possibly reflecting a vestigial trait, inherited from an ancient Nodal-dependent regulation of neurogenesis on the left."*

Answer to comments of reviewer 2:

1. Line 47: I think the authors should delete this non-informational sentence from the Introduction. It doesn't help to make the presented work of greater interest to a broad audience. "From a translational research perspective, it is also important to evaluate the relevance of model organisms for the analysis of neurological disorders in humans."

lines 52-53: We have deleted this sentence as suggested.

2. Line 56: "Habenular territories have also been implicated in human pathological conditions, such as depression or drug addiction and the understanding of these dysregulations, as inferred from analyses in animal models, implies a robust knowledge of cross-species conservations and divergences". I do not think "implies" is a correct word choice here. Do the authors mean requires or necessitates?

Either way, I don't follow the logic. There is no real explanation as to why knowledge of cross-species conservation and divergence is useful for understanding depression or drug addiction in humans. This is an unsubstantiated assertion and, again, statements of justification like these do not help make the study of broader interest. We should just accept the fact that the present study is largely descriptive, of high quality and data-rich, but will likely only be of interest to a specialized audience.

We agree that the understanding of human pathological conditions is not directly related to the knowledge of cross-species conservations and divergences. However, in a context where the zebrafish is used as model to decipher habenula-linked mental disorders, we feel that it is important to stress that a robust understanding of evolutionary conservations and divergences lies behind the notion of animal model. We have therefore rephrased our sentence to clarify our point. As suggested, we have also replaced "implies" by "requires".

lines 58-61: "*Habenular territories have also been implicated in human pathological conditions, such as depression or drug addiction^{7,8}, and knowledge transfers from animal models to humans require a robust knowledge of cross-species conservations and divergences*".

3. This is still a challenging paper to wade through. It might help if the authors refrained from using so many stacked modifiers (e.g., "asymmetric neuronal identity choices of post-mitotic progenitors") We have removed stacked modifiers as suggested:

- line 102-103: "*choice of a right-prevailing neuronal identity*" instead of "*a right-prevailing neuronal identity choice*"
- lines 395-396: "*asymmetric neuronal identities*" instead of "*neuronal identity asymmetries*".
- lines 525-526: "*an ancestral Wnt-dependent genetic program, promoting a right neuronal identity in lateral habenulae*" instead of "*a conserved Wnt-dependent genetic program, promoting an ancestral lateral right habenula neuronal identity*"
- line 550: "*asymmetric neuronal identities*" instead of "*neuronal identity asymmetries*".
- line 561: "*asymmetric modifications of neuronal identities*" instead of "*neuronal identity asymmetric modifications*".

4. Some suggested corrections:

Line 153: “ScSox1 and ScProx1 expressions are confined” change to “ScSox1 and ScProx1 expression patterns are confined”

We have followed this suggestion (line 158).

Line 173: habenular expression not expressions

We have implemented this correction (line 177).

Overuse of “highlight” – use show, demonstrate, reveal, depict, etc.

We have replaced many occurrences of the term "highlight" by more accurate terms throughout the manuscript as suggested.

Change “diversifications” to diversification throughout

This is the only modification, that we would prefer not to take. The plural form indeed insists on our finding that different independent diversifications have taken place in major osteichthyan lineages (tetrapods and the neopterygian group of ray-finned fishes) (lines 210, 573, 578).

I agree with the other reviewer that field-specific terms are (still) not adequately described/explained. For example, the authors do not explain the term cyclostome (e.g., jawless vertebrate) on line 395.

We have checked that this term had been previously defined in the text (line 87).

Line 553: “This difference is not absolute, as an early neurogenetic asymmetry in the zebrafish is Nodal-dependent (in the zebrafish)” Authors should remove the second “in the zebrafish”.

We have implemented this correction (lines 564-565).

Line 556: “In addition, the molecular programs that repress Wnt activity on the left side respectively downstream Nodal in the catshark but under parapineal influences in the zebrafish remain to be established” is an awkward sentence that would benefit from punctuation or re-writing for greater clarity. Anyway, line 565 makes the same point but more clearly so there is no need for stating this twice (“Such variations might in turn lay the ground for major mechanistic transitions, such as the switch from a Nodal-dependent left repression of Wnt signaling, as observed in the catshark, to a parapineal-dependent repression, as described in the zebrafish”).

We have maintained but clarified this sentence.

lines 564-567: *"This difference is not absolute, as an early neurogenetic asymmetry in the zebrafish is Nodal-dependent⁶¹, possibly reflecting a vestigial trait, inherited from an ancient Nodal-dependent regulation of neurogenesis on the left."*